# Scaling Large Language Models with Fully Sparse Activations

**Hongyu Wang**[*]  **Shuming Ma**[*]  **Ruiping Wang**  **Furu Wei**

**Reviewed on OpenReview:** `https://openreview.net/forum?id=MntjMCroiE`

## Abstract

Activation sparsity can reduce the inference cost of large language models (LLMs) by lowering both compute and memory traffic. Yet most existing approaches sparsify only FFN intermediate states, leaving substantial portions of inference effectively dense. We study how to scale fully sparsely activated LLMs, in which every activation participating in linear transformations is sparse. We focus on two questions: how to train such models effectively, and how activation sparsity affects model quality as scale increases. We develop a pre-training recipe that enables effective training fully sparsely activated LLMs from scratch, including using squared ReLU as activation function, top-$K$ sparsification and a straight-through estimator for the remaining linear layers. Extensive experiments spanning model sizes, training-token budgets, and target sparsity levels reveal that its performance gap to dense baselines narrows with model scale, increases nonlinearly with sparsity, while remaining largely insensitive to the training-token budget. Finally, we investigate post-training activation sparsification of pre-trained dense models via both training-free techniques and supervised fine-tuning, and observe a similar trend as pre-training experiments: larger models are more robust to sparsification, and exhibit increasingly sparse activation patterns. Overall, our results provide practical training recipes and empirical guidance for building and scaling LLMs with fully sparse activations.[1]

## 1 Introduction

Large language models (LLMs) have become the workhorse of modern natural language processing (Qwen et al., 2025; Dubey et al., 2024), delivering strong performance across a wide range of tasks and enabling practical systems such as assistants, agents, and retrieval-augmented applications. As LLM capability continues to improve with scale, the dominant barrier to broader deployment is increasingly the inference cost of serving these models at low latency.

Sparsity is a promising lever for reducing inference cost. By inducing zeros in the tensors that participate in inference, sparsity can reduce arithmetic by skipping zero-valued computation and memory traffic by lowering the amount of data moved between memory and compute units. In the LLM setting, most existing activation-sparsity approaches (Song et al., 2024b; Zhang et al., 2024b; Luo et al., 2025) concentrate on sparsifying FFN intermediate states (e.g., the hidden expansion in MLP blocks), largely because these states are produced immediately after non-linear activations (e.g., SiLU) and therefore exhibit a naturally heavy-tailed distribution that makes them particularly amenable to sparsification. However, restricting sparsity to FFN intermediates leaves substantial portions of inference effectively dense, which limits end-to-end efficiency gains. This motivates going beyond "FFN-only" sparsity toward LLMs with fully sparse activations. This shift raises two practical questions:

- How can we design and train an LLM with fully sparse activations to reliably boost model sparsity?

[*]Equal contribution. Hongyu Wang and Ruiping Wang are with the Key Laboratory of AI Safety of Chinese Academy of Sciences (CAS), Institute of Computing Technology, CAS and University of Chinese Academy of Sciences, Beijing. Shuming Ma and Furu Wei are with Microsoft Research. Corresponding author: Hongyu Wang (wanghongyu22@mails.ucas.ac.cn).

[1]The code is available at `https://github.com/ustcwhy/Q-Sparse`.

- How does the degree of activation sparsity affect LLM performance as we scale the model and data?

In this work, we first develop a practical recipe to train fully sparsely-activated LLMs from scratch. We systematically examine the design and optimization choices that make full activation sparsity feasible during pre-training, including the selection of activation/sparsification functions and gradient approximation strategies, where many neurons are inactive and would otherwise receive weak learning signals.

Building on this training recipe, we then investigate the effect of different activation sparsity levels on model performance. By varying target sparsity while scaling model size and training tokens, extensive controlled experiments reveal that **the performance gap between sparsely-activated and dense models narrows with model scale, increases nonlinearly with sparsity, while remaining largely insensitive to the training-token budget.** This suggests that sparse activations become increasingly favorable as models scale, providing a path to reduce inference cost with diminishing quality loss.

Beyond training from scratch, we also study how to sparsify pre-trained dense models. We apply activation sparsification at different target sparsity levels under two practical post-training settings: training-free sparsification and supervised fine-tuning with sparsity constraints. Across extensive experiments on language and multimodal models, we observe results that are consistent with pre-training: **despite being trained with dense activations, larger models remain more robust to aggressive activation sparsification.** Specifically, without any further training, Qwen2.5-32B with 40% activation sparsity only has 0.65% degradation compared to dense baseline on average accuracy. In summary, our contributions are as follows:

- We investigate training fully sparsely activated LLMs, where the activations of every linear transformation are sparse. We find that squared ReLU paired with top-$K$ activation sparsification and a straight-through estimator (STE) for the remaining linear layers provides a strong and reliable recipe for fully sparsely activated LLMs.

- We perform extensive scaling experiments for fully sparsely activated LLMs and observe that, the performance gap between sparsely-activated and dense models narrows with model scale, increases nonlinearly with sparsity, while remaining largely insensitive to the training data scale.

- We study post-training activation sparsification of pretrained dense LLMs and observe a similar trend: despite being trained with dense activations, larger models exhibit increasingly sparse activation patterns, suggesting an emergent tendency toward sparsity with scale.

The paper is organized as follows. In Section 2, we formalize full activation sparsity in LLMs and quantify inference efficiency across a broad range of models under varying sparsity ratios. Section 3 introduces our pre-training recipe for fully sparsely activated LLMs, including the choice of FFN activation function, sparsification strategies for other linear layers, and gradient approximation methods. In Section 4, we systematically study how activation sparsity impacts performance as we scale model size and training-token budgets. Section 5 investigates activation sparsification in post-training settings, including supervised fine-tuning and training-free approaches. Finally, Section 7 discusses structured activation sparsity in LLMs, its compatibility with quantization-aware pre-training, and the integration of sparse linear layers with MoE.

## 2 Preliminary

### 2.1 Activation Sparsity in LLMs

Modern LLMs typically adopt Transformer as the backbone. It consists of stacks of a self-attention layers followed by a feed-forward networks (FFN) layer. The attention layer can be formulated as:

$$Q, K, V = \mathbf{W}_q \text{LN}(X), \mathbf{W}_k \text{LN}(X), \mathbf{W}_v \text{LN}(X)$$
$$\text{MSA}(X) = X + \mathbf{W}_o \text{Attention}(Q, K, V)$$

where $\mathbf{W}_q$, $\mathbf{W}_k$, $\mathbf{W}_v$ and $\mathbf{W}_o$ denote the learnable parameters and have a shape of $\mathbf{R}^{d \times d}$ (without GQA). The computation for FFN layers can be expressed as:

$$\text{Gate} = \mathbf{W}_{\text{up}}(\text{LN}(X)) \cdot \sigma(\mathbf{W}_{\text{gate}}(\text{LN}(X))$$
$$\text{FFN}(X) = X + \mathbf{W}_{\text{down}}\text{Gate}$$

where $\mathbf{W}_{\text{down}}$, $\mathbf{W}_{\text{up}}$, $\mathbf{W}_{\text{gate}}$ have a shape of $\mathbf{R}^{d \times d_f}$, $\mathbf{R}^{d_f \times d}$, $\mathbf{R}^{d_f \times d}$, respectively. $d$ is the hidden dimension, and $d_f$ is the intermediate dimension for FFN. $\sigma$ denote the activation function.

Let $X \in \mathbf{R}^{d \times 1}$, $W \in \mathbf{R}^{d \times d}$ and $Y = WX$ denote input (activations), weight and output of a linear layer, respectively. **Activation sparsity is defined as the proportion of zero entries in the input $X$.**

$$\text{Sparsity}_W = \frac{\sum \mathbf{1}[X_i = 0]}{d} \times 100\% \tag{1}$$

Previous research (Zhang et al., 2024b; Song et al., 2024b) mainly focuses on the sparsity of the intermediate states caused by ReLU-based activation function in $\mathbf{W}_{\text{down}}$ of FFNs. We extend it to all linear layers of LLMs to boost a higher model-wide sparsity. In this work, we mainly investigate the activation sparsity of each linear layers in LLMs, including $\mathbf{W}_{\text{q,k,v,o}}$ in attention layers and $\mathbf{W}_{\text{up,gate,down}}$ in FFN layers.

## 2.2 Real-world Inference Efficiency

In this section, we measure the real-world inference efficiency of fully sparsely-activated LLMs. During inference, sparse activations allows us to first prune the zero entries in activations and corresponding rows in weight. Given the sparsity ratio $S$ in activation $X$, the pruned activation $X_{sp}$ and weight $W_{sp}$ have a shape of $\mathbf{R}^{(1-S)d \times 1}$ and $\mathbf{R}^{d \times (1-S)d}$, respectively. Therefore, it significantly reduces the I/O of weight and computation FLOPs, especially for single-batch scenarios. Following Liu et al. (2025), we benchmark the end-to-end single-batch decoding latency (in token/sec) by integrating top-$K$ sparsification with TEAL's kernel on NVIDIA A6000. We use GPT-Fast's standard inference benchmarking setup. The input length is roughly 5 tokens and the output length is at most 200 tokens.

We evaluate various popular LLM families, including Mistral (Jiang et al., 2023), LLaMA-2 (Touvron et al., 2023), LLaMA-3 (Dubey et al., 2024) and Qwen2.5 (Qwen et al., 2025) models. Activation sparsity was varied across the range [20%, 30%, 40%, 50%, 60%]. We adopt top-$K$ sparsification for all linear layers within LLMs except input and output embeddings. As shown in Table 1, Qwen2.5-72B achieves 44.2%, 67.8%, 99.6% improvement compared to the dense baseline when the sparsity ratio is 30%, 40%, 50% and 60%, respectively. Additionally, since the proportion of latency attributable to linear projections increases with model size, larger models have higher speedup given the same sparsity ratio, which shows that activation sparsity is friendly for model scaling. The results highlight the inference advantages of fully sparsely-activated LLMs, particularly on edge devices.

# 3 Fully Sparsely-Activated Large Language Model

In this section, we investigate the effect of different design dimensions for fully sparsely-activated LLMs, including activation function in FFNs, sparsification function and gradient approximation.

## 3.1 Activation Function

We first investigate the performance of different activation function which mainly affects the activation sparsity before down projection $\mathbf{W}_{\text{down}}$ in FFN layers. Modern LLMs usually adopt SiLU and leave the activations at half-precision (e.g., BF16). We observed that these activations are not naturally sparse despite the distributions are long-tailed and have massive amount of entries around zero.

Table 1: The end-to-end single-batch decoding latency (in token/sec) of various LLMs varying sparsity ratios.

| Models | Dense | Sparsity 20% | Sparsity 30% | Sparsity 40% | Sparsity 50% | Sparsity 60% |
|---|---|---|---|---|---|---|
| Mistral-7B-v0.1 | 43.82 | 44.64 (+1.9%) | 49.63 (+13.3%) | 55.86 (+27.5%) | 63.57 (+45.1%) | 73.68 (+68.1%) |
| LLaMA-2-7B | 46.70 | 47.94 (+2.7%) | 53.25 (+14.0%) | 59.92 (+28.3%) | 68.09 (+45.8%) | 78.87 (+68.9%) |
| LLaMA-2-13B | 24.41 | 26.01 (+6.6%) | 29.14 (+19.4%) | 33.10 (+35.6%) | 38.14 (+56.3%) | 44.93 (+84.1%) |
| LLaMA-2-70B | 4.48 | 5.04 (+12.5%) | 5.66 (+26.3%) | 6.45 (+43.9%) | 7.50 (+67.4%) | 8.94 (+99.5%) |
| Qwen2.5-7B | 44.10 | 44.60 (+1.1%) | 48.72 (+10.5%) | 54.98 (+24.7%) | 62.01 (+40.6%) | 70.93 (+60.8%) |
| Qwen2.5-14B | 22.25 | 23.33 (+4.9%) | 25.91 (+16.5%) | 29.13 (+30.9%) | 32.98 (+48.2%) | 38.36 (+72.4%) |
| Qwen2.5-32B | 8.35 | 9.00 (+7.9%) | 9.87 (+18.3%) | 10.95 (+31.2%) | 12.28 (+47.2%) | 13.93 (+66.9%) |
| Qwen2.5-72B | 4.47 | 5.04 (+12.9%) | 5.66 (+26.7%) | 6.45 (+44.2%) | 7.50 (+67.8%) | 8.92 (+99.6%) |
| LLaMA-3-8B | 41.43 | 42.28 (+2.1%) | 46.71 (+12.7%) | 52.19 (+26.0%) | 58.85 (+42.1%) | 67.43 (+62.8%) |
| LLaMA-3-70B | 4.02 | 4.47 (+11.2%) | 4.95 (+23.0%) | 5.54 (+37.7%) | 6.30 (+56.6%) | 7.29 (+81.1%) |

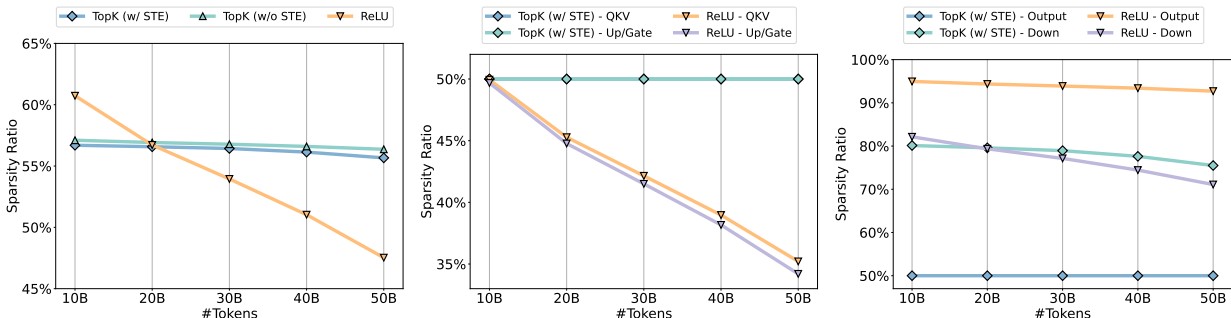

Figure 1: The model-wide sparsity ratio (Left) and each component's sparsity (Middle and Right) of different sparsification functions. As the training progresses, the overall sparsity of ReLU continuously decreases, especially for the inputs to attention (i.e., $\mathbf{W}_{q,k,v}$) and FFN layers (i.e., $\mathbf{W}_{up, gate}$).

We further compare it to different activation function with INT8 quantization, including SiLU, ReLU and squared ReLU. As shown in Table 2, compared to BF16 precision, INT8 quantization has no perplexity and accuracy loss when training from scratch, while enabling 19.4% sparsity in activations of down projections. Using ReLU leads to performance degradation, a drop of 0.02 training loss compared with SiLU function. Similar results

Table 2: The training loss of different activation function in BF16 and INT8. We report the model-wide sparsity and activation sparsity of $\mathbf{W}_{down}$ in FFN layers.

| Method | Precision | Sparsity$^{\text{model}}$ | Sparsity$^{\text{down}}$ | Loss↓ |
|---|---|---|---|---|
| SiLU | BF16 | 0.0% | 0.0% | 3.16 |
| SiLU | INT8 | 10.1% | 19.4% | 3.16 |
| ReLU | INT8 | 21.3% | 72.0% | 3.18 |
| ReLU$^2$ | INT8 | 22.1% | 71.1% | 3.16 |

are also reported by Zhang et al. (2024b). Additionally, squared ReLU matches SiLU on the training loss, while boosting the sparsity level of down projection to over 80%. **Above all, for activation function, using squared ReLU has minimum impact on overall performance and offers high sparsity for activations of $\mathbf{W}_{\text{down}}$ in FFN layers.**

## 3.2 Sparsification Function

We extend activation sparsity to all linear layers of LLMs beyond just the activation function. We investigate the performance and sparsity of ReLU (Mirzadeh et al., 2023) and top-$K$ sparsification for $\mathbf{W}_{q,k,v,o}$ in attention layers and $\mathbf{W}_{up, gate}$ in FFN layers. For top-$K$ sparsification, we mask a proportion (i.e., 1 - $K$) of entries with smaller magnitude for each token $X$. Since ReLU sets the gradient of sparsified entries to zero during backpropagation, we adopted the same estimation method and removed the straight-through estimator (STE) for top-K sparsification to ensure a fair comparison.

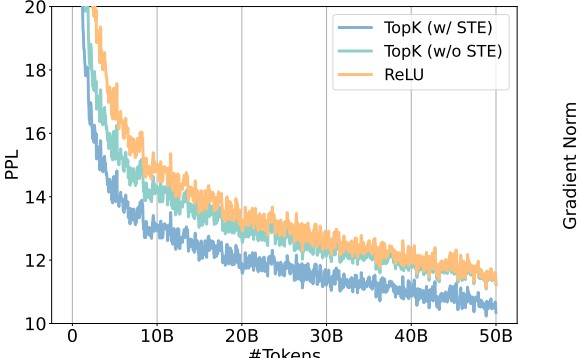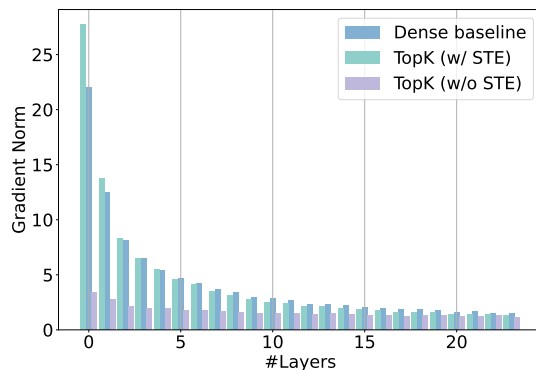

Figure 2: **Left**: the training loss curves of different sparsification functions. All models are trained from scratch with 300M size and 50B tokens. **Right**: the average magnitude of each projection's gradient of dense baseline, sparsely-activated models with and without STE across different layers. The visualization is conducted with 300M model size on a subset of the valid set of C4 (Raffel et al., 2019).

As shown in the left part of Figure 2, we observed that despite top-$K$ (Top-$K$ w/o STE) converges faster than ReLU, it achieves a similar training perplexity at the end of training. However, as for ReLU, the sparsity ratio of $\mathbf{W}_{q,k,v}$ and $\mathbf{W}_{up, gate}$ continuously decreases as the training progresses, while the only activations of $\mathbf{W}_o$ and $\mathbf{W}_{down}$ kept at a high sparsity level (i.e., over 90%). In contrast, the sparsity ratio remains unchanged with the top-K sparsification. We visualize the sparsity of each linear layer throughout training progress in Figure 1. These findings suggest that the models are learned to be dense at inputs to attention and FFN layers, and have much sparse intermediate states. **Above all, for sparsification function, using top-$K$ sparsification achieves a higher sparsity rate over ReLU while maintaining the same performance.**

### 3.3 Gradient Approximation

Most works (Mirzadeh et al., 2023) on training sparsely-activated models use the vanilla back-propagation algorithm to compute the gradient through the sparsity function. They zero the gradients of the non-activated neurons, which hinders learning if these neurons are frequently sparsified across all tokens. A simple solution is to use the straight-through estimator (Bengio et al., 2013) which directly bypasses the gradient through sparsification function without being zeroed-out.

We present the loss curves of top-$K$ sparsification with and without STE in the left part of Figure 2. STE significantly accelerates the convergence of sparsely-activated models and achieves lower training perplexity. We further visualize the average $l2$ norm of each projection's gradient across different layers for dense model, top-$K$ sparsification with and without STE. As shown in the right part of Figure 2, without STE, the gradient is much smaller at the bottom layers, while STE can preserve the magnitude of the gradients. More discussions on STE and visualizations for each components are detailed in Appendix A and Appendix E, respectively. **Above all, directly bypassing the gradients through sparsification significantly improves the convergence speed and overall performance of sparsely-activated models.**

## 4 Pre-training Activation Sparsification

### 4.1 Experimental Setup

To investigate the scaling property of the proposed fully sparsely-activated LLM shown in Section 3, we train a series of language models with different sparsity levels. We adopt squared ReLU as activation function and quantized top-$K$ sparsification for the other linear layers. All activations are per token quantized into 8-bit integers (i.e., $[-128, 127]$) using the absmax function (Dettmers et al., 2022). We deploy STE to

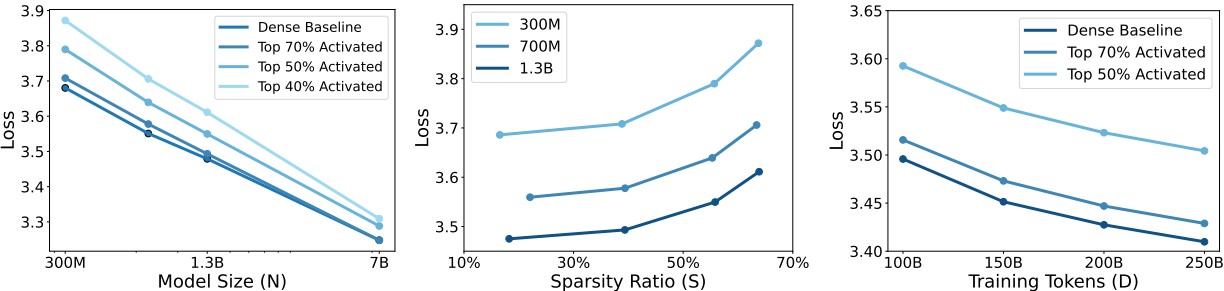

Figure 3: The scaling curves of the sparsely-activated models regrading to the model size $N$ (Left), sparsity ratio $S$ (Middle), and training tokens count $D$ (Right).

conduct gradient approximation. Additionally, we use a standard Transformer++ implementation, including GLU (Shazeer, 2020), RMS normalization (Zhang & Sennrich, 2019), rotary embedding (Su et al., 2024) and removing all bias. We use the Sentencepiece tokenizer from LLaMA to preprocess data. The training data is a subset randomly sampled from Redpajama dataset (TogetherAI, 2023). We report these models' perplexity on the validation set of C4 (Raffel et al., 2019). More details can be found in the Appendix D.

## 4.2 Main Results

**Diminishing Gap as Model Size $N$ Scales.** We study pre-training with model parameters $N$ and activations in various sparsity levels $S$. As shown in Figure 3, we found that with a fixed sparsity level $S$, as the number of parameters $N$ increases, the performance gap between sparsely-activated models and dense baselines diminishes. For instance, at 30% activation sparsity, although there remains a gap in 300M models, when the total model size is scaled to 7B, the perplexity of sparsely-activated LLMs fully match that of the dense baselines with the same parameter count.

To investigate the ability of in-context learning for fully sparsely-activated and dense LLMs, we evaluate the zero-shot accuracy of these models on a range of language tasks using normalized log probs., including ARC-Easy (Yadav et al., 2019), ARC-Challenge (Yadav et al., 2019), Hellaswag (Zellers et al., 2019), PIQA (Bisk et al., 2019) and LAMBADA (Paperno et al., 2016). All models are trained with 50 billion tokens from Redpajama for a fair comparison. We adopt squared ReLU as activation function and use top-$K$ sparsification with STE for the other linear layers except input/output embedding. $K$ is varied from $[100\%, 70\%, 50\%, 40\%]$.

Table 3 summarizes the results of fully sparsely-activated LLMs across various model size and activation sparsity ratios. Given a fixed sparsity ratio, the accuracy on the end tasks continuously increases, as the total model size grows. Furthermore, we observed that as the parameter count scales, the performance gap between sparsely-activated and dense models decreases. In the 300 million scale, the dense model exceeds the sparse model with 60% sparsity ratio by 2.5% in average accuracy, whereas this difference decreases to 1.9% at the 7 billion scale. Additionally, we observed that given the similar active model size, a large sparsely-activated model outperforms a small dense model. For example, an 1.3B model with 50% sparsity ratio outperforms the dense 700M model by a gain of 0.72% average accuracy, and a 700M model with 60% sparsity ratio achieves an improvement of 0.43% average accuracy. These results proves the effectiveness of fully sparsely-activated models given the same inference FLOPs. We provide our theoretical understanding of this trend in Appendix C.

**Constant Gap as Training Data $D$ Scales.** We investigate LLM pre-training under varying training-token budgets $D$ and activation sparsity ratios $S$. To emulate an over-training regime under constrained compute, we fix the model size at $N$=700M parameters and sweep $D \in \{100, 150, 200, 250\}$B tokens together with $S \in \{0\%, 30\%, 50\%\}$. This design yields token-to-parameter ratios $D/N$ of up to 350, and thus provides a stress test for how activation sparsity behaves when data is plentiful relative to capacity. As shown in the right panel of Figure 3, as $D$ increases, the relative ordering of the curves remains stable: dense models consistently achieve the lowest loss, followed by 30% and 50% sparsity, and the performance gaps between

Table 3: The zero-shot accuracy of fully sparsely-activated LLMs across various model size and activation sparsity levels.

| Sparsity | Size | ARC-Challenge | ARC-Easy | Hellaswag | LAMBADA | PIQA | Average | Δ |
|----------|------|---------------|----------|-----------|---------|------|---------|---|
| 0% | 300M | 23.29 | 45.24 | 41.08 | 45.41 | 65.13 | 44.030 | +0.000 |
| 30% | 300M | 24.32 | 43.60 | 40.76 | 46.75 | 66.16 | 44.318 | +0.288 |
| 50% | 300M | 24.40 | 43.10 | 39.21 | 43.24 | 63.82 | 42.754 | -1.276 |
| 60% | 300M | 24.15 | 41.58 | 37.26 | 40.17 | 64.47 | 41.526 | -2.504 |
| 0% | 700M | 25.34 | 46.76 | 44.68 | 49.89 | 67.90 | 46.914 | +0.000 |
| 30% | 700M | 26.45 | 47.14 | 43.79 | 50.07 | 67.25 | 46.940 | +0.026 |
| 50% | 700M | 25.94 | 45.66 | 43.15 | 46.46 | 66.65 | 45.572 | -1.342 |
| 60% | 700M | 26.71 | 43.64 | 41.61 | 44.28 | 66.05 | 44.458 | -2.456 |
| 0% | 1.3B | 28.41 | 47.56 | 47.72 | 52.14 | 69.10 | 48.986 | +0.000 |
| 30% | 1.3B | 26.54 | 47.18 | 47.06 | 52.44 | 69.42 | 48.528 | -0.458 |
| 50% | 1.3B | 27.13 | 46.04 | 45.79 | 51.41 | 67.79 | 47.632 | -1.354 |
| 60% | 1.3B | 26.19 | 46.17 | 44.12 | 49.21 | 66.10 | 46.358 | -2.628 |
| 0% | 7B | 30.97 | 55.93 | 57.24 | 60.10 | 72.20 | 55.288 | +0.000 |
| 30% | 7B | 31.40 | 54.80 | 57.80 | 59.42 | 72.47 | 55.178 | -0.110 |
| 50% | 7B | 30.63 | 55.09 | 56.02 | 59.23 | 71.76 | 54.546 | -0.742 |
| 60% | 7B | 28.75 | 53.11 | 55.63 | 57.52 | 71.82 | 53.366 | -1.922 |

sparsity ratios remain approximately constant over the entire range of token budgets. This suggests that, at fixed model size, activation sparsity primarily induces a roughly additive penalty in loss rather than altering the scaling exponent, even in an over-training setting.

**Increasing Gap as Sparsity Ratio $S$ scales.** We further examine how activation sparsity impacts pre-training when the model size and training-token budget are held fixed. Concretely, we fix the data size to $D$=50B tokens and sweep the model size $N \in \{300M, 700M, 1.3B\}$ while varying the activation sparsity ratio over $S \in \{0\%, 30\%, 40\%, 60\%\}$. As shown in the middle panel of Figure 3, increasing $S$ consistently degrades the final validation loss across all model sizes, and the degradation becomes more pronounced as sparsity grows. Importantly, the effect is not linear in $S$: moderate sparsity (e.g., 30%) incurs a relatively small penalty, whereas pushing sparsity further (e.g., 40% to 60%) leads to a noticeably larger gap relative to the dense baseline. This trend is consistent across the three model scales we consider, suggesting that while sparse activation can be applied broadly, there is a practical trade-off: higher sparsity ratios offer greater potential efficiency benefits but come with increasingly larger accuracy costs under a fixed data budget. We attribute this behavior to the use of the straight-through estimator (STE) for gradient approximation. While STE can partially mitigate weakened gradients for neurons that are frequently masked by sparsification, under aggressive activation sparsity, it introduces a increasingly biased (and potentially incorrect) gradient signal during backpropagation, which can misalign the optimization trajectory and hinder convergence. Overall, these results highlight that the sparsity ratio acts as a primary knob governing the accuracy-efficiency frontier in fully sparsely activated pre-training.

## 5 Post-training Activation Sparsification

Top-$K$ activation sparsification can also be used for off-the-shelf dense models. We investigate the training-free sparsification for pre-trained dense models in Section 5.1 and sparsification in supervised fine-tuning in Section 5.2.

### 5.1 Training-free Sparsification

We choose Qwen2.5-series models, since it supports various model size. We directly apply top-$K$ sparsification for all linear layers except input/output embedding. The sparsification is training-free and does not require any calibration data. We perform a grid sweep over combinations of total parameter count $N \in \{3, 7, 14\}$ billion and activation sparsity $S \in \{0\%, 20\%, 30\%, 40\%, 50\%, 60\%\}$. We evaluate the validation perplexity of these models on C4. As shown in the left part of Figure 4, for a fixed model size $N$ and token budget $D$,

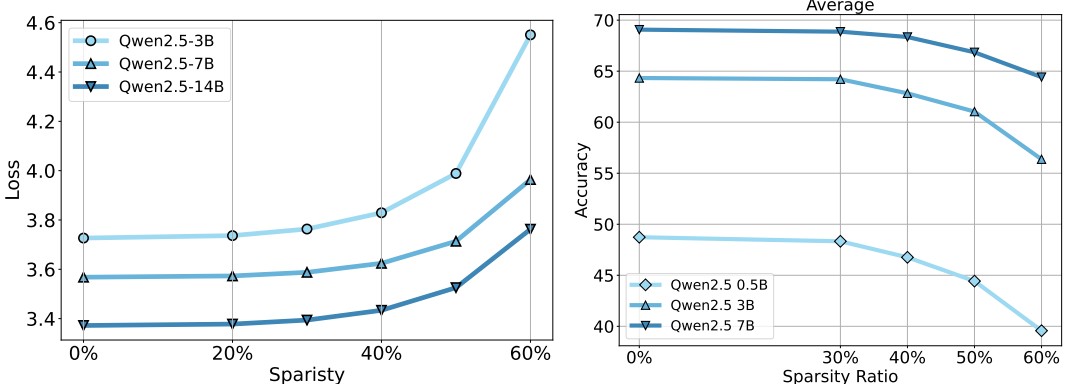

Figure 4: **Left:** the validation perplexity on C4 of Qwen2.5 models varying different sparsity ratios. **Right:** the average accuracy of fine-tuned Qwen2.5 models varying different sparsity ratios across 15 benchmarks.

Table 4: Performance of Qwen2.5 models under different sparsity levels across various benchmarks.

| Sparsity | ARC-C | HellaSwag | PIQA | Winogrande | MMLU | GSM-8K | MBPP | CSQA | Avg. |
|---|---|---|---|---|---|---|---|---|---|
| *Qwen2.5-7B* | | | | | | | | | |
| 0% | 51.6 | 79.0 | 79.8 | 73.1 | 71.9 | 82.8 | 64.2 | 85.4 | 73.47 |
| 20% | 51.6 | 78.8 | 79.8 | 72.9 | 71.5 | 83.4 | 64.4 | 85.0 | 73.43 |
| 30% | 51.5 | 78.4 | 79.3 | 71.1 | 71.1 | 80.9 | 62.6 | 84.0 | 72.36 |
| 40% | 52.0 | 77.4 | 79.0 | 71.2 | 69.7 | 78.8 | 61.6 | 83.3 | 71.63 |
| 50% | 50.3 | 74.9 | 78.5 | 71.0 | 67.7 | 71.6 | 53.8 | 80.3 | 68.51 |
| 60% | 49.2 | 69.0 | 77.0 | 65.9 | 61.5 | 54.2 | 43.2 | 72.6 | 61.58 |
| *Qwen2.5-32B* | | | | | | | | | |
| 0% | 55.7 | 84.1 | 82.3 | 75.3 | 80.8 | 89.8 | 73.4 | 88.4 | 78.73 |
| 20% | 56.5 | 84.1 | 82.4 | 76.5 | 80.8 | 89.4 | 73.2 | 88.6 | 78.94 |
| 30% | 56.0 | 83.8 | 82.8 | 76.6 | 80.5 | 90.1 | 74.0 | 87.9 | 78.96 |
| 40% | 55.8 | 83.2 | 82.3 | 75.1 | 79.7 | 88.9 | 71.8 | 87.8 | 78.08 |
| 50% | 55.3 | 81.6 | 81.1 | 75.3 | 78.3 | 87.0 | 67.2 | 86.2 | 76.50 |
| 60% | 53.9 | 78.1 | 79.8 | 73.4 | 74.1 | 80.7 | 61.8 | 81.5 | 72.91 |

moderate activation sparsity has little to no impact on loss, whereas pushing $S$ to higher levels leads to a clear and increasingly noticeable degradation in performance, consistent with the qualitative trend predicted by our analysis.

Furthermore, we evaluate Qwen2.5-7B and Qwen2.5-32B under varying activation sparsity. We report results on ARC-Challenge (ARC-C, 0-shot), HellaSwag (0-shot), PIQA (0-shot), Winogrande (0-shot), MMLU (0-shot), GSM8K (5-shot), MBPP (3-shot), and CommonsenseQA (CSQA, 0-shot). As shown in Table 4, we find that for the same sparsity ratio, the performance gap between sparse and dense models shrinks as model size increases. For example, at 40% activation sparsity, Qwen2.5-7B drops by 1.84% in average accuracy relative to its dense baseline, whereas Qwen2.5-32B incurs only a 0.65% drop. Overall, these results suggest that larger LLMs are more robust to activation sparsification, consistent with our scaling observations.

For multimodal models, we consider the instruction-tuned Qwen2-VL models at the 2B and 7B scales, and vary the activation sparsity ratio over $S \in \{0\%, 30\%, 40\%, 50\%, 60\%\}$. We evaluate zero-shot performance on MMMU (val) (Yue et al., 2024), MMStar (Chen et al., 2024), SeedBench-2-Plus (Li et al., 2024), AI2D (Kembhavi et al., 2016), ChartQA (Masry et al., 2022), InfoVQA (val) (Mathew et al., 2022), and DocVQA (val) (Mathew et al., 2021). To ensure a controlled and fair comparison, we use the LMM-Eval toolkit (Zhang et al., 2024a) with a consistent evaluation protocol across model sizes and sparsity settings. As shown in Table 5, for a fixed sparsity ratio $S$, the performance gap between sparsely activated and dense

Table 5: Performance of Qwen2-VL models under different sparsity levels across various benchmarks.

| Sparsity | MMMU | MMStar | SeedBench[2+] | AI2D | ChartQA | InfoVQA | DocVQA | Avg. | Δ |
|---|---|---|---|---|---|---|---|---|---|
| *Qwen2-VL-2B-Instruct* | | | | | | | | | |
| 0% | 39.3 | 42.5 | 61.9 | 69.8 | 70.6 | 63.2 | 89.2 | 62.36 | +0.00 |
| 30% | 38.3 | 41.6 | 61.4 | 68.7 | 70.7 | 62.6 | 88.8 | 61.73 | -0.63 |
| 40% | 37.3 | 42.4 | 61.0 | 67.2 | 69.9 | 60.7 | 88.5 | 61.00 | -1.36 |
| 50% | 35.0 | 40.7 | 55.6 | 64.0 | 64.9 | 56.9 | 85.2 | 57.47 | -4.89 |
| 60% | 33.1 | 35.7 | 47.3 | 51.2 | 47.4 | 40.5 | 74.5 | 47.10 | -15.26 |
| *Qwen2-VL-7B-Instruct* | | | | | | | | | |
| 0% | 49.9 | 55.7 | 69.0 | 79.8 | 80.6 | 74.7 | 93.8 | 71.93 | +0.00 |
| 30% | 50.9 | 55.7 | 67.9 | 79.6 | 80.5 | 74.5 | 93.8 | 71.84 | -0.09 |
| 40% | 50.8 | 55.2 | 67.8 | 79.1 | 80.9 | 73.6 | 93.8 | 71.60 | -0.33 |
| 50% | 45.9 | 53.0 | 66.5 | 77.7 | 79.2 | 72.0 | 93.2 | 69.64 | -2.29 |
| 60% | 44.1 | 49.4 | 64.1 | 74.6 | 75.4 | 69.8 | 91.6 | 67.00 | -4.93 |

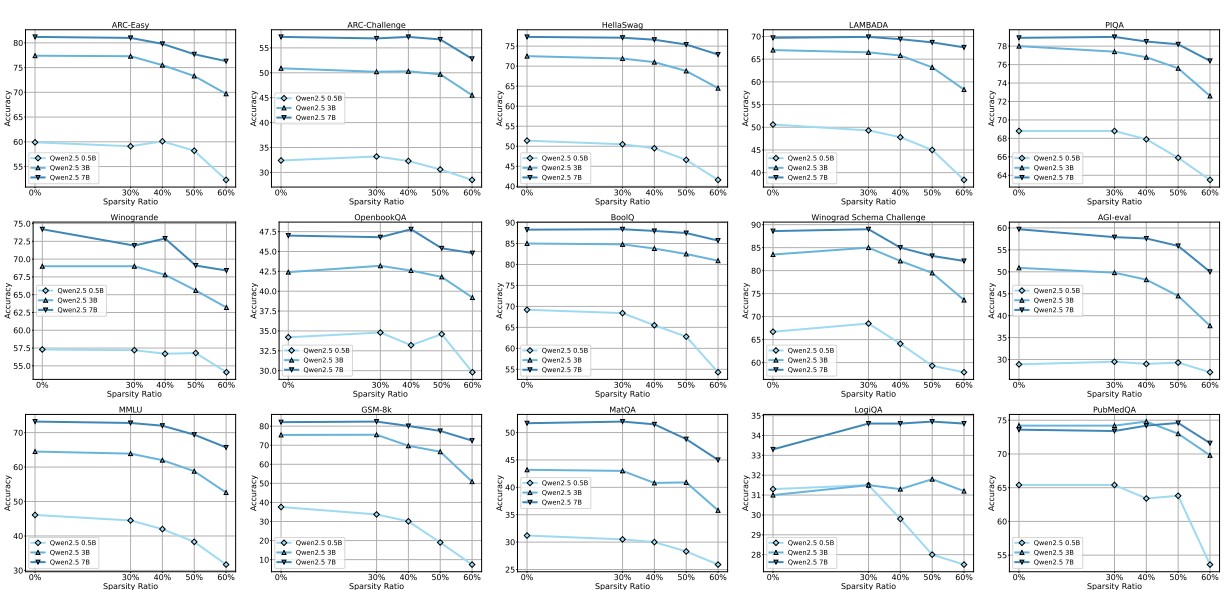

Figure 5: The accuracy of fine-tuned Qwen2.5 models varying different sparsity ratios at 0.5B, 3B and 7B scales on various benchmarks.

models shrinks markedly as model size increases. In particular, at $S$=60%, Qwen2-VL-2B suffers a 15.26% absolute accuracy drop relative to its dense baseline, whereas the 7B model drops by only 4.93%. This trend mirrors our findings for text-only LLMs and suggests that larger multimodal models are substantially more tolerant to aggressive activation sparsification.

## 5.2 Supervised Fine-tuning

For supervised fine-tuning, we fine-tune the base model of Qwen2.5 at 0.5B, 3B and 7B scales on the OpenOrca (Lian et al., 2023) and MetaMathQA (Yu et al., 2023) dataset using LLaMA-Factory (Zheng et al., 2024). We select the subset generated by GPT-4 for OpenOrca dataset. All models are trained with one epoch and a peak learning rate of 5e-6. The batch size is set as 128. We disable the dropout and weight decay during training. We apply top-$K$ sparsification for all linear layers and use STE to conduct gradient approximation. The sparsity ratio is varied from $\{0\%, 30\%, 40\%, 50\%, 60\%\}$.

We report the zero-shot accuracy of these models on ARC-easy, ARC-challenge, HellaSwag, LAMBADA, PIQA, Winogrande, OpenbookQA (Mihaylov et al., 2018), BoolQ (Clark et al., 2019), Winograd Schema Challenge (Levesque et al., 2012), AGI-eval (Zhong et al., 2024), MathQA (Amini et al., 2019), LogiQA (Liu

Table 6: The validation perplexity on C4 and zero-shot accuracy of end tasks for top-$K$ and block top-$K$ sparsification varying different model size.

| Method | Size | PPL↓ | ARC-Challenge↑ | ARC-Easy↑ | Hellaswag↑ | LAMBADA↑ | PIQA↑ | Avg.↑ |
|--------|------|------|----------------|-----------|------------|----------|-------|-------|
| Block Top-$K$ | 300M | 14.11 | 23.98 | 42.34 | 38.35 | 41.86 | 64.53 | 42.21 |
| Top-$K$ | | 13.84 | 24.40 | 43.10 | 39.21 | 43.24 | 63.82 | 42.75 |
| Block Top-$K$ | 700M | 12.67 | 26.37 | 45.37 | 42.54 | 46.48 | 66.70 | 45.49 |
| Top-$K$ | | 12.45 | 25.94 | 45.66 | 43.15 | 46.46 | 66.65 | 45.57 |

et al., 2020), PubMedQA (Jin et al., 2019); and five-shot accuracy of MMLU (Hendrycks et al., 2021) and GSM-8K (Cobbe et al., 2021).

We report the average accuracy for the three model scales under varying activation sparsity ratios in the right panel of Figure 4. Overall, the fine-tuned models exhibit a clear and consistent dependence on $S$: performance is largely preserved at moderate sparsity, while more aggressive sparsification leads to increasingly noticeable degradation. In addition, for a fixed $S$, the gap between sparsely activated and dense models narrows as the model scale increases, mirroring the trend we observe when training base models from scratch. Detailed per-task results are provided in Figure 5.

## 6 Discussion

### 6.1 Structured Activation Sparsification

While top-$K$ sparsification is effective in single-batch mode, it is less compatible with batched inference on current GPU architectures. Correspondingly, most prior work on activation sparsity (Liu et al., 2023; 2025; Song et al., 2023) has primarily evaluated the single-batch regime (e.g., batch size = 1). To ensure fair comparison and to follow this common protocol, our main experiments adopt the same setting and therefore focus on unstructured top-$K$ sparsification. Zhou et al. (2021); Lin et al. (2023) nonetheless suggest that structured sparsity (e.g., N:M sparsity, where $N$ out of $M$ consecutive elements are zero) is more hardware-friendly and can enable efficient batched execution with optimized GPU kernels. To better align with such constraints, we also consider block-level top-$K$ sparsification: we partition activations into blocks of size $M$ and apply top-$K$ independently within each block, yielding $(1 - K)M$ zeros per block. Standard top-$K$ is recovered as a special case when the block size equals the hidden dimension.

We train 300M and 700M models with top-$K$ and block top-$K$ sparsification. 1.3B models' results can be found in Appendix B. We use squared ReLU as activation function and quantize the inputs to 8-bit integers using absmax function. For backward, we adopt STE to bypass the gradient through top-$K$ or block top-$K$ function. The block size is set as 32, which is recommended by the previous work (Lin et al., 2023) on N:M sparse kernels. The sparse ratio is set as 50% for a fair comparison. All models are trained with 50 billion tokens from Redpajama dataset. We evaluate the perplexity on valid set of C4, and zero-shot accuracy on the end tasks following the setup shown in Section 4.2. As shown in Table 6, block top-$K$ sparsification achieves comparable perplexity and accuracy on the end tasks compared to top-$K$ sparsification in both 300M and 700M size. We further report the results of training-free sparsification for Qwen2.5-7/32B in Appendix B. in We hope these findings will encourage further research on structured activation sparsity for batched inference on both system and algorithm level.

### 6.2 Compatibility to Quantization-aware Pre-training

As the model size grows, the limited memory bandwidth required for transferring model weights becomes a major bottleneck, especially at the decoding stage for LLMs. Model quantization serves as a promising approach to reduce the memory footprint. Therefore, we investigate the scaling law of sparsely-activated LLMs with 1.58-bit pre-training (Wang et al., 2023; Ma et al., 2024).

We train a series of BitNet b1.58 models of various scales and sparsity ratios with 50 billion tokens following the experimental setup shown in Section 4.1. We sweep the model size $N \in \{300M, 700M, 1.3B, 7B\}$ while

Table 7: The zero-shot accuracy of fully sparsely-activated LLMs trained with 1.58-bit weights across various model size and activation sparsity levels.

| Sparsity | Size | ARC-Challenge | ARC-Easy | Hellaswag | LAMBADA | PIQA | Average | Δ |
|----------|------|---------------|----------|-----------|---------|------|---------|---|
| 0% | 300M | 22.78 | 41.12 | 36.52 | 42.79 | 64.25 | 41.492 | +0.000 |
| 30% | 300M | 22.01 | 40.91 | 36.19 | 41.43 | 63.71 | 40.850 | -0.642 |
| 50% | 300M | 22.18 | 41.20 | 34.73 | 36.37 | 62.95 | 39.486 | -2.006 |
| 60% | 300M | 22.53 | 40.24 | 32.94 | 32.47 | 61.86 | 38.008 | -3.484 |
| 0% | 700M | 24.15 | 45.50 | 42.57 | 48.71 | 66.81 | 45.548 | +0.000 |
| 30% | 700M | 24.74 | 44.91 | 41.98 | 47.25 | 66.54 | 45.084 | -0.464 |
| 50% | 700M | 24.40 | 43.64 | 40.71 | 44.34 | 64.80 | 43.578 | -1.970 |
| 60% | 700M | 24.57 | 41.71 | 39.15 | 41.55 | 63.71 | 42.138 | -3.410 |
| 0% | 1.3B | 25.94 | 49.66 | 46.60 | 51.85 | 68.12 | 48.434 | +0.000 |
| 30% | 1.3B | 27.82 | 46.30 | 46.36 | 51.12 | 68.39 | 47.998 | -0.436 |
| 50% | 1.3B | 25.77 | 45.96 | 44.95 | 48.46 | 67.14 | 46.456 | -1.978 |
| 60% | 1.3B | 24.83 | 45.37 | 43.61 | 47.99 | 66.32 | 45.624 | -2.810 |
| 0% | 7B | 30.63 | 55.98 | 57.17 | 59.60 | 72.58 | 55.192 | +0.000 |
| 30% | 7B | 29.61 | 55.18 | 56.44 | 59.27 | 72.09 | 54.518 | -0.674 |
| 50% | 7B | 30.20 | 52.15 | 55.49 | 58.35 | 71.33 | 53.504 | -1.688 |
| 60% | 7B | 30.29 | 51.94 | 53.81 | 56.01 | 71.00 | 52.610 | -2.582 |

Table 8: Performance of OLMoE-1B-7B-0125 and Qwen1.5-MoE-A2.7B models under different sparsity levels across various benchmarks.

| Sparsity | ARC-C | HellaSwag | PIQA | Winogrande | MMLU | GSM-8K | MBPP | CSQA | Avg. |
|----------|-------|-----------|------|------------|------|--------|------|------|------|
| *OLMoE-1B-7B-0125* | | | | | | | | | |
| 0% | 49.2 | 78.2 | 79.8 | 68.8 | 53.4 | 52.8 | 21.8 | 52.3 | 57.04 |
| 20% | 49.3 | 78.0 | 79.4 | 69.2 | 53.1 | 51.9 | 20.4 | 50.8 | 56.51 |
| 30% | 49.0 | 77.3 | 79.2 | 68.0 | 52.3 | 50.3 | 21.4 | 48.8 | 55.79 |
| 40% | 48.7 | 76.2 | 78.6 | 65.8 | 50.3 | 46.6 | 20.2 | 47.3 | 54.21 |
| 50% | 46.2 | 73.1 | 76.9 | 65.7 | 47.4 | 35.5 | 15.2 | 43.7 | 50.46 |
| 60% | 42.6 | 66.9 | 75.5 | 62.6 | 40.4 | 17.1 | 8.0 | 39.1 | 44.03 |
| *Qwen1.5-MoE-A2.7B* | | | | | | | | | |
| 0% | 44.5 | 77.3 | 80.2 | 69.1 | 60.9 | 60.9 | 37.8 | 80.2 | 63.86 |
| 20% | 44.7 | 77.0 | 79.9 | 68.3 | 60.7 | 59.9 | 37.2 | 79.2 | 63.36 |
| 30% | 43.6 | 76.6 | 79.6 | 68.1 | 59.6 | 60.5 | 37.8 | 79.4 | 63.15 |
| 40% | 42.5 | 75.0 | 78.8 | 67.8 | 58.4 | 56.4 | 36.4 | 75.8 | 61.39 |
| 50% | 41.5 | 72.1 | 78.5 | 65.4 | 55.2 | 46.7 | 29.4 | 70.8 | 57.45 |
| 60% | 37.5 | 65.7 | 75.5 | 65.0 | 46.4 | 20.5 | 18.2 | 57.4 | 48.28 |

varying the activation sparsity ratio over $S \in \{0\%, 30\%, 50\%, 60\%\}$. We evaluate the zero-shot accuracy of these models following the setup shown in Section 4.2. As show in Table 7. The sparsely-activated BitNet b1.58 models with 30% sparsity achieves similar performance than the dense baselines while offering the lower inference compute budget (i.e., active parameters). We observed that, like in full-precision pre-training, larger models exhibit increased activation sparsity in 1.58-bit LLMs, resulting in a reduced performance gap between sparsely-activated and dense models. These findings indicate that activation sparsity complements 1-bit pre-training, and their combination can enhance model performance during inference.

## 6.3 Fully Sparsely-activated MoE

Mixture-of-Experts (MoE) models (Lepikhin et al., 2021) are widely used to scale model capacity efficiently via expert-level sparsity: each token is routed to a small subset of experts, so only a fraction of parameters are activated per forward pass. In contrast, our approach targets activation-level sparsity by enforcing fine-grained sparsity within the linear transformations applied to every token. These two forms of sparsity operate at different granularities and are therefore largely orthogonal: MoE reduces the number of experts (modules) executed, while our method reduces the number of active hidden dimensions within each executed

module. As a result, they are naturally complementary and could be combined to further improve inference efficiency of LLMs.

As an initial probe, Table 8 reports the effect of training-free activation sparsification on two MoE LLMs, OLMoE-1B-7B-0125 and Qwen1.5-MoE-A2.7B, evaluated on ARC-Challenge (0-shot), HellaSwag (0-shot), PIQA (0-shot), Winogrande (0-shot), MMLU (0-shot), GSM-8K (5-shot), MBPP (3-shot), and Common-senseQA (0-shot). These results suggest that, while MoE routing already induces conditional computation at the expert level, additional fine-grained activation sparsity can still be imposed post hoc. We leave the systematic study and pre-training of fully sparsely activated MoE models to future work.

## 7   Related Work

**Activation sparsification.**   The inputs to linear layers in LLMs often exhibit long-tailed magnitude distributions, suggesting that only a small fraction of hidden dimensions are strongly activated for a given token. This motivates activation sparsification as a principled way to reduce inference cost while preserving accuracy. DejaVu (Liu et al., 2023) shows that activation sparsity is prevalent in LLMs and can be predicted with lightweight predictors, enabling conditional computation without expensive gating. Mirzadeh et al. (2023) demonstrates that replacing the commonly used SiLU with ReLU yields negligible degradation in convergence and final performance, while reducing compute and memory traffic. They further inserts ReLU before each linear projection to increase end-to-end sparsity. PowerInfer (Song et al., 2023) exploits sparsity in the FFN down-projection to build a GPU-CPU hybrid inference engine: frequently activated neurons are kept on GPU, while infrequently activated neurons are computed on CPU, reducing GPU memory pressure and CPU-GPU transfers. TurboSparse (Song et al., 2024b) proposes the dReLU activation to improve the accuracy-sparsity trade-off, and ProSparse (Song et al., 2024a) introduces progressive sparsity regularization to gradually increase sparsity during training and stabilize optimization. In contrast to most prior work that primarily targets sparsifying FFN intermediate states, we extend activation sparsity to all linear transformations in the LLM and systematically study a pre-training recipe for fully sparsely activated LLMs, including activation choices and sparsification/optimization strategies.

**Scaling LLMs.**   A large body of work studies how different scaling factors affect LLM performance and efficiency, spanning parameter count and data budgets (Hoffmann et al., 2022), architectural choices (Tay et al., 2023; Ludziejewski et al., 2024), numerical precision (Dettmers & Zettlemoyer, 2023; Kumar et al., 2024; Ouyang et al., 2024), and data composition (Liu et al., 2024). Within this line, MoE scaling has received particular attention: recent studies examine how the number of experts, routing policies, and routing granularity shape the accuracy-efficiency frontier (Clark et al., 2022; Ludziejewski et al., 2025). Luo et al. (2025) provides an empirical scaling study of activation sparsity, introducing a performance-aware PPL-$p\%$ sparsity metric and reporting systematic trends of sparsity with training data, activation functions (e.g., SiLU vs. ReLU), and architectural shape, offering quantitative guidance for designing LLMs with higher activation sparsity. Our work is complementary to these efforts: we focus on how activation sparsity of linear transformations interacts with model scale and training regime, and how this interaction informs both pre-training and post-training sparsification.

## 8   Conclusion

In this work, we studied how to scale fully sparsely-activated LLMs, where every activation participating in linear transformations is sparse. We developed a practical pre-training recipe that enables training fully sparsely activated LLMs from scratch. Extensive scaling experiments show that the performance gap between sparsely-activated and dense models narrows with model scale, increases nonlinearly with sparsity, while remaining largely insensitive to the training data scale. These results indicates that larger models are increasingly robust to full activation sparsification. Finally, we investigated post-training activation sparsification of pretrained dense models using both training-free methods and supervised fine-tuning, and found trends that mirror those in pre-training. Taken together, our findings provide both actionable training recipes and empirical guidance for building and scaling LLMs with fully sparse activations.

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

## A    Discussion on STE

We further investigate the effect of the straight-through estimator (STE) on training sparse models from scratch. All models in this study contain 1.3B parameters and are trained on 50B tokens from the RedPajama dataset. We consider three sparsity ratios, 40%, 60%, and 80%, and report the C4 validation perplexity, where lower values indicate better performance.

Table 9: Comparison of STE and w/o STE under different sparsity settings.

| Method | Sparsity 40% | Sparsity 60% | Sparsity 80% |
|---|---|---|---|
| w/o STE | 12.65 | 13.82 | 14.41 |
| STE | 11.47 | 12.28 | 15.15 |

As shown in Table 9, STE improves optimization at moderate sparsity levels. At 40% sparsity, STE reduces the C4 perplexity from 12.65 to 11.47, corresponding to an absolute improvement of 1.18. At 60% sparsity, the improvement is even larger, reducing perplexity from 13.82 to 12.28. These results suggest that STE provides a useful gradient approximation for the discrete sparsification operation when the active parameter budget remains sufficiently large.

However, the benefit of STE does not persist under very high sparsity. At 80% sparsity, STE increases the perplexity from 14.41 to 15.15, indicating worse final performance. This reversal is consistent with our hypothesis that the gradient approximation bias introduced by STE becomes more pronounced as sparsity becomes more aggressive. When only a small fraction of parameters remains active, the mismatch between the forward sparsification pattern and the backward gradient approximation may hurt training stability and lead to suboptimal convergence.

## B    Structured Activation Sparsity

In this section, we provide additional experiments on structured activation sparsity. While the main paper focuses on unstructured Top-$K$ activation sparsity, structured sparsification is also practically relevant, since block-wise sparsity can better align with hardware-friendly execution patterns and reduce the overhead associated with fully unstructured activation selection.

First, beyond the 300M and 700M-scale comparisons reported in the main paper, we compare block Top-$K$ sparsity with unstructured Top-$K$ sparsity at the 1.3B scale under 50% sparsity. Both models are trained from scratch on 50B tokens. As shown in Table 10, when sparse models are trained from scratch, block Top-$K$ performs comparably to unstructured Top-$K$ at the same sparsity level. At 50% sparsity, block Top-$K$ obtains a C4 perplexity of 12.03, close to 11.79 for unstructured Top-$K$. The average downstream accuracy is also similar: 47.13% for block Top-$K$ and 47.45% for Top-$K$. These results suggest that the structural constraint introduced by block-wise sparsification does not fundamentally limit model quality when the sparsity pattern is incorporated during pre-training. With sufficient training, the model can adapt its representations and optimization dynamics to the imposed block structure, leading to performance close to that of unstructured activation sparsity.

The training-free setting provides a more challenging test, since sparsification is directly applied to a pretrained dense model without additional adaptation. We evaluate training-free activation sparsification on Qwen2.5-7B and Qwen2.5-32B, using block sizes of 32 and 128 and sparsity ratios from 20% to 60%. The dense model is included as the 0% sparsity reference. As shown in Tables 11 and 12, block Top-$K$ remains close to unstructured Top-$K$ under relatively mild sparsity. For example, on Qwen2.5-7B, the average scores of Top-$K$,

Table 10: The validation perplexity on C4 and zero-shot accuracy of end tasks for top-$K$ and block top-$K$ sparsification with 1.3B model size.

| Method | Size | PPL↓ | ARC-Easy↑ | ARC-Challenge↑ | HellaSwag↑ | LAMBADA↑ | PIQA↑ | Winogrande↑ | Avg.↑ |
|---|---|---|---|---|---|---|---|---|---|
| Block TopK | 1.3B | 12.03 | 45.5 | 26.2 | 44.5 | 47.6 | 66.9 | 52.1 | 47.13 |
| TopK | | 11.79 | 47.0 | 25.5 | 45.1 | 47.9 | 67.2 | 52.0 | 47.45 |

block Top-$K$ with block size 32, and block Top-$K$ with block size 128 are 73.43%, 73.14%, and 73.11% at 20% sparsity, respectively. Similarly, on Qwen2.5-32B, the corresponding average scores are 78.94%, 78.56%, and 78.75%. This indicates that, when the sparsity ratio is low, the additional block-wise constraint has limited impact because most activations are still retained.

The effect of block size becomes more visible as sparsity increases. A larger block size generally provides more flexibility for selecting important activations within each block, and therefore better approximates unstructured Top-$K$. For instance, at 60% sparsity on Qwen2.5-32B, block Top-$K$ with block size 128 achieves an average score of 71.40%, which is much closer to unstructured Top-$K$ at 72.91% than block Top-$K$ with block size 32 at 66.73%. These results suggest that training-free block sparsification is most effective when the sparsity level is moderate or when the block size is sufficiently large, as both conditions reduce the restriction imposed by the structured selection pattern.

Importantly, the structured sparsification results are consistent with the main scaling trend observed in the paper: larger models are more robust to activation sparsity. Thus, even when the sparsification pattern is structured, increasing model scale substantially improves tolerance to activation sparsity.

Table 11: Performance of Top-$K$ and Block Top-$K$ training-free activation sparsification with Qwen2.5-7B.

| Method | Sparsity | ARC-C | HellaSwag | PIQA | Winogrande | MMLU | GSM-8K | MBPP | CSQA | Avg. |
|---|---|---|---|---|---|---|---|---|---|---|
| Dense | 0% | 51.6 | 79.0 | 79.8 | 73.1 | 71.9 | 82.8 | 64.2 | 85.4 | 73.47 |
| TopK | 20% | 51.6 | 78.8 | 79.8 | 72.9 | 71.5 | 83.4 | 64.4 | 85.0 | 73.43 |
| Block TopK (size=32) | 20% | 51.1 | 78.6 | 79.5 | 73.6 | 71.4 | 82.3 | 63.8 | 84.8 | 73.14 |
| Block TopK (size=128) | 20% | 51.2 | 78.7 | 79.5 | 72.9 | 71.3 | 82.9 | 63.6 | 84.8 | 73.11 |
| TopK | 30% | 51.5 | 78.4 | 79.3 | 71.1 | 71.1 | 80.9 | 62.6 | 84.0 | 72.36 |
| Block TopK (size=32) | 30% | 51.0 | 77.8 | 78.6 | 72.3 | 70.6 | 79.5 | 63.2 | 84.0 | 72.13 |
| Block TopK (size=128) | 30% | 51.0 | 78.0 | 78.7 | 72.7 | 70.5 | 81.3 | 62.8 | 84.1 | 72.39 |
| TopK | 40% | 52.0 | 77.4 | 79.0 | 71.2 | 69.7 | 78.8 | 61.6 | 83.3 | 71.63 |
| Block TopK (size=32) | 40% | 50.7 | 76.2 | 78.6 | 70.8 | 68.8 | 77.5 | 59.8 | 81.7 | 70.51 |
| Block TopK (size=128) | 40% | 50.4 | 76.6 | 78.8 | 71.0 | 69.4 | 76.1 | 58.6 | 81.4 | 70.29 |
| TopK | 50% | 50.3 | 74.9 | 78.5 | 71.0 | 67.7 | 71.6 | 53.8 | 80.3 | 68.51 |
| Block TopK (size=32) | 50% | 48.5 | 73.1 | 76.9 | 67.2 | 65.1 | 68.5 | 52.2 | 77.8 | 66.16 |
| Block TopK (size=128) | 50% | 48.3 | 73.8 | 78.0 | 67.8 | 65.9 | 68.5 | 53.2 | 80.0 | 66.94 |
| TopK | 60% | 49.2 | 69.0 | 77.0 | 65.9 | 61.5 | 54.2 | 43.2 | 72.6 | 61.58 |
| Block TopK (size=32) | 60% | 40.0 | 60.6 | 71.9 | 61.8 | 51.7 | 30.6 | 24.2 | 60.4 | 50.15 |
| Block TopK (size=128) | 60% | 44.2 | 65.8 | 74.2 | 61.5 | 57.8 | 45.9 | 33.6 | 66.3 | 56.16 |

## C  Discussion on Constant Scaling Exponent $\alpha$

Our current hypothesis is that activation sparsity mainly changes the scaling factor but not the model-size scaling exponent. We start from the sparsity-aware scaling law:

$$L(N, S) = E + \frac{A(S)}{N^{\alpha(S)}},$$

where $N$ and $S$ denote the model size and activation sparsity ratio, respectively. Since higher activation sparsity generally degrades performance, the scaling factor $A(S)$ is strictly positive and increases with $S$. Empirically, small changes in $S$ induce only modest and smooth variations in the loss. To capture this, we assume that $L$ is globally Lipschitz-continuous with respect to $S$, i.e., there exists a constant $K > 0$, independent of $N$, such that

Table 12: Performance of Top-$K$ and Block Top-$K$ training-free activation sparsification with Qwen2.5-32B.

| Method | Sparsity | ARC-C | HellaSwag | PIQA | Winogrande | MMLU | GSM-8K | MBPP | CSQA | Avg. |
|---|---|---|---|---|---|---|---|---|---|---|
| Dense | 0% | 55.7 | 84.1 | 82.3 | 75.3 | 80.8 | 89.8 | 73.4 | 88.4 | 78.73 |
| TopK | 20% | 56.5 | 84.1 | 82.4 | 76.5 | 80.8 | 89.4 | 73.2 | 88.6 | 78.94 |
| Block TopK (size=32) | 20% | 57.0 | 84.0 | 82.1 | 74.2 | 80.7 | 89.8 | 72.8 | 87.9 | 78.56 |
| Block TopK (size=128) | 20% | 56.0 | 84.1 | 82.1 | 76.1 | 80.6 | 89.0 | 73.6 | 88.5 | 78.75 |
| TopK | 30% | 56.0 | 83.8 | 82.8 | 76.6 | 80.5 | 90.1 | 74.0 | 87.9 | 78.96 |
| Block TopK (size=32) | 30% | 56.6 | 83.6 | 82.0 | 75.5 | 80.4 | 90.4 | 74.6 | 87.6 | 78.84 |
| Block TopK (size=128) | 30% | 55.7 | 83.5 | 82.2 | 76.4 | 80.3 | 88.8 | 71.6 | 87.6 | 78.26 |
| TopK | 40% | 55.8 | 83.2 | 82.3 | 75.1 | 79.7 | 88.9 | 71.8 | 87.8 | 78.08 |
| Block TopK (size=32) | 40% | 55.1 | 82.5 | 81.2 | 75.6 | 79.4 | 89.5 | 70.4 | 86.7 | 77.55 |
| Block TopK (size=128) | 40% | 55.5 | 82.9 | 82.3 | 74.7 | 79.3 | 89.7 | 70.6 | 87.0 | 77.75 |
| TopK | 50% | 55.3 | 81.6 | 81.1 | 75.3 | 78.3 | 87.0 | 67.2 | 86.2 | 76.50 |
| Block TopK (size=32) | 50% | 54.0 | 80.9 | 81.6 | 73.3 | 77.2 | 86.1 | 68.0 | 83.6 | 75.59 |
| Block TopK (size=128) | 50% | 53.0 | 81.2 | 81.6 | 75.5 | 77.5 | 86.1 | 69.4 | 84.4 | 76.09 |
| TopK | 60% | 53.9 | 78.1 | 79.8 | 73.4 | 74.1 | 80.7 | 61.8 | 81.5 | 72.91 |
| Block TopK (size=32) | 60% | 50.3 | 73.3 | 76.8 | 70.8 | 68.5 | 71.1 | 47.4 | 75.6 | 66.73 |
| Block TopK (size=128) | 60% | 52.7 | 76.5 | 79.0 | 73.5 | 71.9 | 77.9 | 61.2 | 78.5 | 71.40 |

$$\left| \frac{\partial L}{\partial S} \right| \le K, \quad \forall N, S.$$

If we differentiate it with respect to $S$, we obtain

$$\frac{\partial L}{\partial S} = \frac{A'(S)}{N^{\alpha(S)}} - \frac{A(S)\alpha'(S)\ln N}{N^{\alpha(S)}}.$$

If $\alpha'(S) \neq 0$, then for $S = S_0$ and $N \to \infty$, the term involving $\ln N$ diverges, violating the uniform Lipschitz bound independent of $N$. Hence, $\alpha'(S) = 0$, which implies that $\alpha(S)$ is constant. Intuitively, this means that sparsity introduces an additional penalty term, but does not fundamentally change how performance improves with scale. Larger models are therefore more robust to activation sparsity because they still benefit from the same scaling exponent, while sparsity mainly shifts the scaling curve through $A(S)$.

## D    Hyperparameters

Table 13 summarizes the detailed hyper-parameters for scaling experiments. For BitNet b1.58 models, we adopt the two-stage learning rate and weight decay scheduling, which is recommended by Ma et al. (2024) for better performance. We disable the dropout and set the gradient clipping as 2.0.

We present the model configuration of BitNet b1.58 and LLaMA LLM models in Table 14. For simplicity, we do not adopt grouped query attention for all models. The experiments were conducted on the equivalent of 128 NVIDIA H100 GPU cards.

Table 13: Hyper-parameters for the scaling experiments of fully sparsely-activated BitNet b1.58 and LLaMA LLM. For data scaling experiments, we use a batch size of 1M tokens.

| Model | Size | Learning Rate | Weight Decay | Batch Size | Adam $\beta$ |
|---|---|---|---|---|---|
| BitNet b1.58 | 300M | $1.8 \times 10^{-3} \to 1.5 \times 10^{-3}$ | $0.1 \to 0$ | 0.5M | (0.9, 0.95) |
| | 700M | $1.5 \times 10^{-3} \to 1 \times 10^{-3}$ | $0.1 \to 0$ | 0.5M | (0.9, 0.95) |
| | 1.3B | $1.2 \times 10^{-3} \to 8 \times 10^{-4}$ | $0.1 \to 0$ | 0.5M | (0.9, 0.95) |
| | 7B | $1 \times 10^{-3} \to 6 \times 10^{-4}$ | $0.1 \to 0$ | 0.5M | (0.9, 0.95) |
| LLaMA LLM | 300M | $6.0 \times 10^{-4}$ | 0.1 | 0.5M | (0.9, 0.95) |
| | 700M | $2.5 \times 10^{-4}$ | 0.1 | 0.5M* | (0.9, 0.95) |
| | 1.3B | $2.0 \times 10^{-4}$ | 0.1 | 0.5M | (0.9, 0.95) |
| | 7B | $1.5 \times 10^{-4}$ | 0.1 | 0.5M | (0.9, 0.95) |

Table 14: Model configurations for the scaling experiments of both BitNet b1.58 and LLaMA LLM.

| Size | Hidden Size | GLU Size | #Heads | #Layers | Seq Length |
|---|---|---|---|---|---|
| 300M | 1024 | 2730 | 16 | 24 | 2048 |
| 700M | 1536 | 4096 | 24 | 24 | 2048 |
| 1.3B | 2048 | 5460 | 32 | 24 | 2048 |
| 7B | 4096 | 11008 | 32 | 32 | 2048 |

# E   More Visualization

We present the gradient's magnitude of each component for the dense baseline, the fully sparsely-activated models trained with and without STE. As shown in Figure 6, STE significantly eases the issue of gradient vanishing, especially at the bottom of the layers.

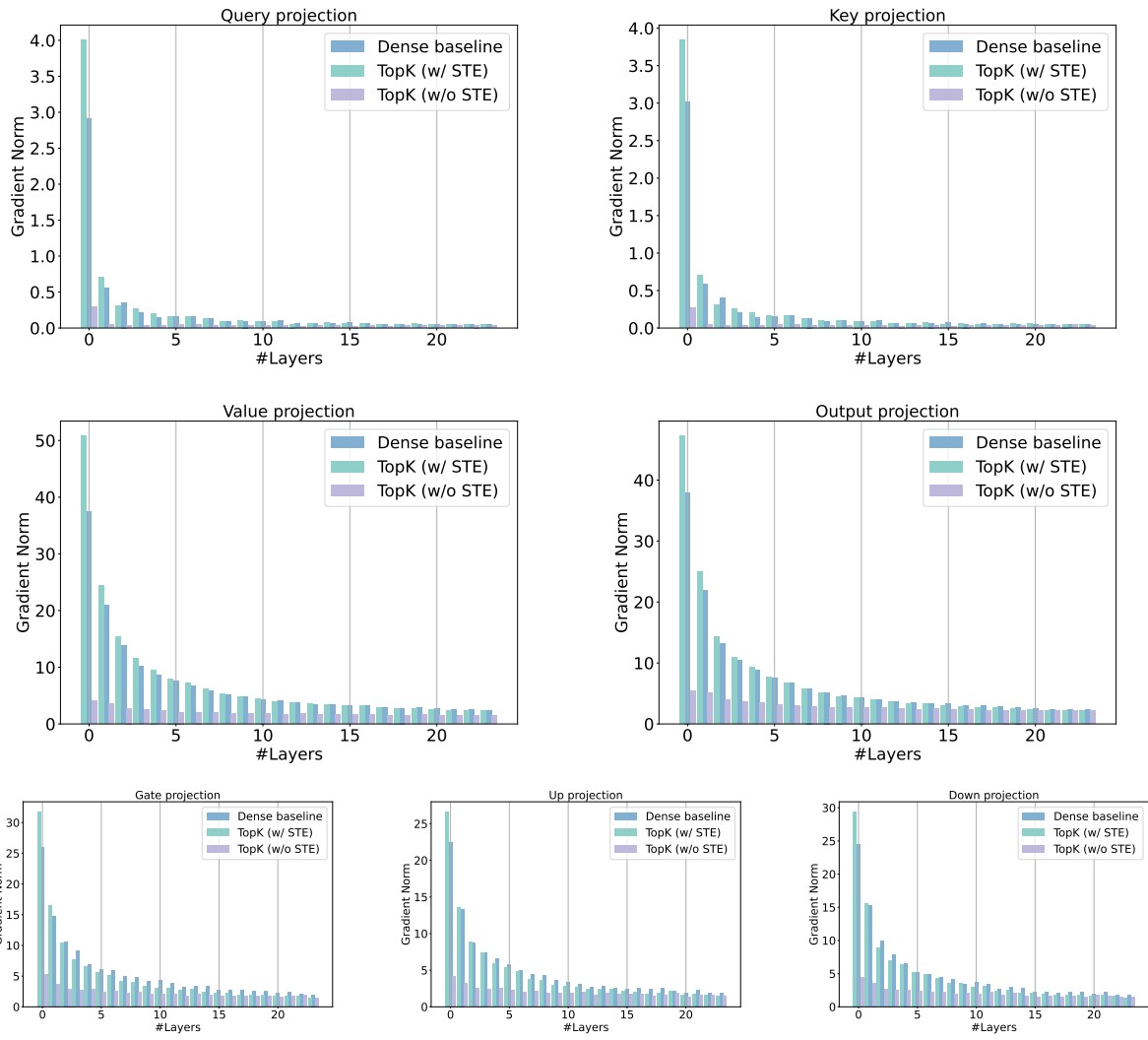

Figure 6: The gradient magnitude of each linear projection of dense baseline, Q-Sparse with and without STE estimator across different layers.

