# OpenReview forum: "Scaling Large Language Models with Fully Sparse Activations"
_TMLR — Accepted by TMLR_

### Review · Reviewer_MEno · 2026-02-23

**Summary Of Contributions:**

This paper studies fully sparsely activated LLMs where Top-K sparsification is applied to all linear layers. The authors propose a training recipe using Squared ReLU and STE, and conduct scaling experiments across model sizes and sparsity levels. They show that larger models are more robust to sparsity and that sparsity mainly introduces a constant performance gap without changing the scaling trend.

Strengths:
- The empirical study is systematic and well controlled across model and data scales.

- The training recipe is practical and clearly described.

- The scaling observations (e.g., constant loss gap, better tolerance in larger models) are interesting and potentially useful.

Weaknesses:
- The paper provides little theoretical insight into why sparsity preserves the scaling exponent.

- The role and bias of STE are not analyzed in depth despite being central to training.

- The practical inference impact is unclear: no wall-clock latency or throughput results are reported, so it is hard to judge whether FLOP reductions translate to real speedups on modern hardware.

**Audience:**

Yes

**Audience Explanation:**

Researchers working on efficient training, sparse models, and scaling laws would likely find the results useful.

**Broader Impact Concerns:**

I do not see major ethical concerns specific to this work.

**Claims And Evidence:**

Yes

**Claims Explanation:**

The claims are mostly supported in the paper.

The empirical claims are supported by systematic experiments across model scales and sparsity levels. The trends reported (e.g., constant performance gap, increased robustness at larger scales) are consistent and presented. However, some claims regarding practical efficiency would benefit from direct wall-clock or hardware-level measurements rather than relying solely on theoretical FLOP reductions.

**Requested Changes:**

- Clarify the practical inference implications. Provide either wall-clock latency / throughput benchmarks or clearly state that the efficiency claims are theoretical (FLOP-based) and may not directly translate to speedups on current hardware.

- Expand the discussion on the role of STE, especially its limitations at high sparsity levels.

- Provide more discussion or hypotheses explaining why sparsity preserves the scaling exponent.

---

> ### Author Response · Authors · 2026-03-23
>
> We thank the reviewer for the careful evaluation and valuable feedback, which helped us improve the paper. We address the reviewer’s concerns point by point below.
>
> **Q1:** Clarify the practical inference implications. Provide either wall-clock latency / throughput benchmarks or clearly state that the efficiency claims are theoretical (FLOP-based) and may not directly translate to speedups on current hardware.
>
> **A1:** We agree that the practical deployment scope should be stated more clearly. Our main contribution is to study **how to train fully sparsely-activated LLMs and how their performance scales with model size, token budget, and sparsity ratio**, rather than to claim universal speedups in all inference settings. Following prior activation-sparsity work, **we already report end-to-end single-batch decoding latency in Table 1 for Mistral, LLaMA-2, LLaMA-3, and Qwen2.5 across multiple sparsity levels.** We will revise the paper to make it more explicit that these speedups may not directly transfer to high-throughput batched inference on current hardware.
>
> **Q2:** Expand the discussion on the role of STE, especially its limitations at high sparsity levels.
>
> **A2:** Thank you for this helpful suggestion. We agree that the role of STE, especially at high sparsity, deserves more discussion. To better address this point, we have added new experiments comparing **with-STE vs. without-STE** training. All models are trained from scratch with 1.3B parameters on 50B tokens from RedPajama dataset, with sparsity ratios of **40%, 60%, and 80%**, and evaluated using **C4 perplexity**.
>
> The validation perplexity of dense baseline on C4 is 11.14.
> ||Sparsity 40%|Sparsity 60%|Sparsity 80%|
> |---|---:|---:|---:|
> |w/o STE|12.65|13.82|**14.41**|
> |STE|**11.47**|**12.28**|15.15|
>
> These results suggest that STE is helpful at moderate sparsity levels (e.g., smaller than 60%), where it substantially improves optimization quality, but its benefit becomes reverses under very high sparsity (e.g., 80%). This is consistent with our hypothesis that, as sparsity becomes more aggressive, the gradient approximation bias introduced by STE becomes larger and can eventually hurt training stability and final performance. We will include these results in the appendix and expand the discussion of STE’s limitations at high sparsity levels in the revision.
>
> **Q3:** Provide more discussion or hypotheses explaining why sparsity preserves the scaling exponent.
>
> **A3:** Our current hypothesis is that activation sparsity mainly changes the **scaling factor** but not the **model-size scaling exponent**. Specifically, for a fixed sparsity ratio $S$, we write the sparsity-aware scaling law as
> $$
> L(N,S)=E+\frac{A(S)}{N^{\alpha(S)}}.
> $$
> If we differentiate it with respect to $S$, we obtain
> $$
> \frac{\partial L}{\partial S} = \frac{A'(S)}{N^{\alpha(S)}} - \frac{A(S)\alpha'(S)\ln N}{N^{\alpha(S)}}.
> $$
> Our key assumption is that, for any model size $N$, the loss changes **smoothly** with sparsity ratio $S$, i.e., $L(N,S)$ is Lipschitz-continuous with respect to $S$. Under this assumption, if $\alpha'(S)\neq 0$, then the second term contains a $\ln N$ factor and would grow with model size, violating the uniform smoothness assumption. Therefore, $\alpha'(S)=0$, which implies that the scaling exponent is approximately **constant** across sparsity levels, and sparsity mainly affects the coefficient $A(S)$.
>
> Intuitively, this means that sparsity introduces an additional penalty term, but does not fundamentally change how performance improves with scale. Larger models are therefore more robust to activation sparsity because they still benefit from the same scaling exponent, while sparsity mainly shifts the scaling curve through $A(S)$. We will clarify this hypothesis in the revision and include the derivation in the appendix.

---

> ### Author Response · Authors · 2026-04-07
>
> Dear Reviewer MEno,
>
> I hope you are doing well.
>
> I am writing to follow up on the discussion, as there is about one week remaining before the discussion period ends, and I have not yet received any further response. I wanted to kindly check whether you might have any additional questions, comments, or concerns that I could help address.
>
> I would be very happy to provide any further clarification if needed. Thank you again for your time and consideration.
>
> Best regards,
>
> Authors.

---

> > ### Comment · Reviewer_MEno · 2026-04-13
> >
> > Thank you for the detailed rebuttal. Your responses were helpful and addressed my main concerns to a satisfactory extent. I appreciate the clearer scoping of the practical efficiency claims, the additional STE ablation, and the added hypothesis regarding the scaling exponent.

---

> > > ### Author Response · Authors · 2026-04-13
> > >
> > > We are delighted that our responses have adequately addressed your concerns. Thank you very much for the time and effort you devoted to reviewing our work. Your valuable suggestions have played an important role in strengthening the quality of our manuscript.

---

### Review · Reviewer_3Yir · 2026-03-06

**Summary Of Contributions:**

This paper studies fully sparse activations in LLMs, meaning sparsity is applied not only to FFN intermediate activations but to every activation that participates in linear transformations. The paper makes three main contributions. First, it proposes a practical recipe for training fully sparsely activated LLMs from scratch, combining squared ReLU in FFNs, top-$K$ sparsification for the remaining linear layers, and a straight-through estimator (STE) during backpropagation. Second, it presents a scaling study showing that the quality gap between sparse and dense models narrows as model size increases, while the penalty grows nonlinearly with higher sparsity levels and appears relatively stable across training-token budgets. Third, it studies post-training activation sparsification for pretrained dense models and reports similar trends for both text-only and multimodal models, namely that larger models are more robust to sparsification. The paper is interesting because it moves beyond FFN-only sparsity and attempts to treat activation sparsity as a model-wide design principle rather than a local trick.

**Strengths**
1. **The paper addresses an important efficiency problem for LLMs:**
Reducing inference cost without giving up much model quality is highly relevant, and the paper tackles this through fully sparse activations rather than restricting sparsity to FFN intermediates only. That broader framing makes the work more ambitious and potentially more impactful than many prior activation-sparsity papers.

2. **The proposed recipe is clear and practically useful:**
The combination of squared ReLU, top-$K$ sparsification, and STE is easy to understand and appears to be well motivated by the empirical ablations. In particular, the paper shows that squared ReLU preserves loss better than ReLU while inducing much higher sparsity, and that STE helps mitigate optimization difficulties such as weak gradient flow.

3. **The experimental scope is broad:**
A major strength is that the paper does not rely on one narrow experimental setting. It studies pre-training from scratch, scaling with model size, scaling with token budget, post-training sparsification, supervised fine-tuning, multimodal models, and some discussion of structured sparsity and MoE extensions. That breadth makes the central trend feel much more credible.

4. **The central empirical finding is interesting and meaningful:**
The observation that the quality gap between sparse and dense models shrinks as model size increases is the most compelling takeaway in the paper. The reported results suggest that larger models are substantially more robust to full activation sparsification, both in pre-training and in post-training settings.

5. **The paper provides evidence of practical relevance, not just perplexity improvements:**
In addition to language-model loss, the paper reports downstream zero-shot results and includes single-batch decoding measurements, which helps connect the work to actual inference scenarios.

6. **The work is positioned well relative to prior literature:**
The paper clearly explains how it differs from prior work focused mainly on FFN sparsity and frames itself as complementary to existing scaling-law and sparsity studies.

**Weaknesses**

1. **The practical systems story is narrower than the paper’s broader efficiency framing may suggest:**
The real-world latency results are mainly for single-batch decoding, and the paper itself notes that unstructured top-$K$ sparsification is less compatible with batched inference on current GPUs. That limitation is important, because many deployment settings care more about throughput and batching than about the specific single-sequence scenario emphasized here.

2. **The paper is stronger empirically than mechanistically:**
It convincingly shows that larger models are more robust to sparsification, but gives less insight into why this happens. The explanation remains somewhat high-level, so the paper reads more as a strong empirical study and recipe paper than as one offering deep theoretical understanding.

3. **Some comparisons would benefit from clearer fairness/accounting:**
The paper discusses similar active model size and inference FLOPs, but the comparison framework could be made more systematic. In particular, it would help to more clearly separate training cost, active inference compute, parameter count, and real wall-clock efficiency, especially when comparing larger sparse models against smaller dense ones.

4. **Structured sparsity is only explored in a limited way:**
Since structured sparsity is more hardware-friendly, it would be useful to see a somewhat broader structured-sparsity evaluation. In the current version, that part feels more like an encouraging add-on than a fully developed experimental component.

5. **The evidence for some practical claims is not equally strong across all settings:**
The paper’s strongest evidence supports the scaling trend and the feasibility of the training recipe. The broader implication that this will readily translate to general deployment gains is less fully established, especially outside the tested inference configuration.

6. **Presentation could be tightened in a few places:**
There are some wording and grammar issues, and a few claims would benefit from more careful phrasing. This does not undermine the paper’s main contribution, but polishing would improve clarity and make the argument more precise.

**Additional Comments:**

I found the paper interesting and generally well executed. The most compelling part is not just the proposed recipe itself, but the consistency of the reported scaling trend across pre-training, post-training sparsification, supervised fine-tuning, and multimodal settings. I also appreciated that the paper includes discussion beyond the main result, such as structured sparsity, low-bit training compatibility, and MoE compatibility, since those make the work feel more forward-looking and practically relevant.

My main reservation is that the paper is strongest as an empirical study plus recipe paper, and somewhat less strong as a systems paper or as a paper offering deep theoretical understanding. If the authors sharpen the practical claims, clarify the comparison protocol, and better articulate the limitations, I think this would be a solid contribution.

**Audience:**

Yes

**Audience Explanation:**

Yes. I expect this paper to interest readers working on efficient LLM inference, sparse computation, scaling laws, post-training compression, and hardware-aware model design. One appealing aspect is that the paper does not only introduce another sparsification heuristic; it tries to answer a broader question about how activation sparsity behaves as model scale increases, and it does so across both pre-training and post-training settings. That makes the paper relevant not just to systems-oriented readers, but also to people interested in optimization, scaling behavior, and efficient foundation models more broadly.

**Broader Impact Concerns:**

I do not see major new ethical concerns beyond those already associated with making LLM inference cheaper and more deployable. In fact, methods that reduce inference cost can have positive impact through lower energy use and broader accessibility. The main concern is that improved inference efficiency can also lower the cost of scaling up deployment of powerful language and multimodal models, including for low-quality or harmful uses. I do not think this is a reason to reject the work, but a brief broader-impact discussion acknowledging both the efficiency benefits and the dual-use nature of improved deployment efficiency would be appropriate.

**Claims And Evidence:**

Yes

**Claims Explanation:**

Overall, the empirical evidence is fairly strong for the paper’s central claim that larger models tolerate full activation sparsification better than smaller ones. The paper supports this with several complementary experiments: controlled pre-training studies across model sizes and sparsity levels, evaluations under varying token budgets, post-training sparsification on pretrained Qwen models, supervised fine-tuning experiments, and additional studies on multimodal models, structured sparsification, quantization-aware pre-training, and MoE models. The main trends are consistent across these settings: moderate sparsity often causes only small degradation, larger models exhibit smaller relative drops than smaller ones, and aggressive sparsity produces increasingly larger quality loss.

That said, I would not say all claims are equally convincing. The evidence is strongest for the existence of the scaling trend and for the usefulness of the proposed training recipe. It is somewhat less strong for broader practical deployment claims, because the real latency study is centered on single-batch decoding with a particular sparse kernel and GPU setup, while the paper itself acknowledges that unstructured top-$K$ sparsification is less compatible with batched inference on current hardware. So the paper clearly demonstrates scientific value and promising practical potential, but some of the broader systems implications should be stated a bit more carefully.

**Requested Changes:**

1. **Clarify the practical deployment scope of the method:**
This is important for acceptance. The paper reports attractive decoding-speed improvements in the single-batch setting, but it also notes that unstructured top-$K$ sparsification is less compatible with batched inference on existing GPUs. The paper should more explicitly separate the scientific claim (“full activation sparsity scales better in larger models”) from the systems claim (“this yields broad real-world speedups”), and discuss where the latter currently does and does not apply.

2. **Improve the discussion of fairness in the compute/quality comparison:**
This is important for acceptance. The paper often motivates sparse models through reduced active computation, and it gives examples where a larger sparse model outperforms a smaller dense model at similar active FLOPs. However, the presentation would be stronger with a clearer and more systematic accounting of training compute, inference FLOPs, active parameters, and wall-clock efficiency, especially when comparing sparse and dense models across scales.

3. **Strengthen the explanation of why the scaling trend occurs:**
This would strengthen the paper. The empirical trend is clear, but the mechanistic explanation remains somewhat limited. The paper briefly attributes degradation at high sparsity to STE bias, but a deeper interpretation of why larger models are more robust to activation sparsity would improve the contribution. Even a more careful hypothesis section or targeted ablation around gradient bias, activation distributions, or layerwise sensitivity would help.

4. **Report variance or statistical stability where feasible:**
This would strengthen the paper. Most results are presented as single curves or tables. Given that several reported differences are fairly small at moderate sparsity, especially around 20–30%, confidence intervals, multiple seeds, or at least a short discussion of run-to-run variability would make the conclusions more convincing.

5. **Expand the structured sparsity discussion:**
This would strengthen the paper. The block top-$K$ experiment is useful, but currently limited to 300M and 700M models at one sparsity setting. Since hardware compatibility is one of the main practical motivations, a slightly broader structured-sparsity study would make the paper more complete.

6. **Tighten a few overstatements and wording issues:**
This would strengthen the paper. There are a few places where the language could be made more precise, for example around practical efficiency, robustness, and the strength of causal explanations. I also noticed a few grammatical issues and minor presentation roughness that should be cleaned up in revision.

---

> ### Author Response · Authors · 2026-03-23
> **Part1**
>
> We sincerely thank the reviewer for the thoughtful comments and helpful suggestions. We provide a point-by-point response to the reviewer’s comments below.
>
> **Q1:** Clarify the practical deployment scope of the method.
>
> **A1:** Our core contribution is to study how to design fully sparsely-activated LLMs and how their performance scales with model size, token budget, and sparsity ratio, rather than to claim universal speedups in all deployment settings. Following prior work [1], the latency results (see Table 1 in our paper) are intended to demonstrate practical gains in the single-batch decoding regime. We will make this scope more explicit and clarify that unstructured top-K sparsity is currently less suitable for batched inference on existing GPUs.
>
> **Q2:** Improve the discussion of fairness in the compute/quality comparison.
>
> **A2:** We will clarify this more systematically in the appendix of next revision. For fairness, our sparse models and dense baselines are trained with the same total parameter count and token budget. At inference time, the sparse models use only a fraction of activations, so their active parameters and inference FLOPs are reduced proportionally to the retained activation ratio. In addition, following prior work [1], we already report end-to-end decoding latency in Table 1 rather than only idealized FLOPs.
>
> **Q3:** Strengthen the explanation of why the scaling trend occurs.
>
> **A3:** Our current hypothesis is that activation sparsity mainly changes the **scaling factor** but not the **model-size scaling exponent**. Specifically, for a fixed sparsity ratio $S$, we write the sparsity-aware scaling law as
> $$
> L(N,S)=E+\frac{A(S)}{N^{\alpha(S)}}.
> $$
> If we differentiate it with respect to $S$, we obtain
> $$
> \frac{\partial L}{\partial S} = \frac{A'(S)}{N^{\alpha(S)}} - \frac{A(S)\alpha'(S)\ln N}{N^{\alpha(S)}}.
> $$
> Our key assumption is that, for any model size $N$, the loss changes **smoothly** with sparsity ratio $S$, i.e., $L(N,S)$ is Lipschitz-continuous with respect to $S$. Under this assumption, if $\alpha'(S)\neq 0$, then the second term contains a $\ln N$ factor and would grow with model size, violating the uniform smoothness assumption. Therefore, $\alpha'(S)=0$, which implies that the scaling exponent is approximately **constant** across sparsity levels, and sparsity mainly affects the coefficient $A(S)$.
>
> Intuitively, this means that sparsity introduces an additional penalty term, but does not fundamentally change how performance improves with scale. Larger models are therefore more robust to activation sparsity because they still benefit from the same scaling exponent, while sparsity mainly shifts the scaling curve through $A(S)$. We will clarify this hypothesis in the revision and include the derivation in the appendix.
>
> **Q4:** Report variance or statistical stability where feasible.
>
> **A4:** Since these models are trained on large token budgets, we expect the main trends to be relatively robust. Our current presentation emphasizes consistency across many tasks and settings rather than repeated-seed reporting. Specifically, in pre-training we evaluate on **5 representative language tasks**; in post-training we evaluate on **8 tasks covering language modeling, math, and code**; and in multimodal experiments we evaluate on **7 tasks**. We also include **perplexity evaluation**, which is often more sensitive to small pre-training differences.

---

> ### Author Response · Authors · 2026-03-23
> **Part2**
>
> **Q5:** Expand the structured sparsity discussion.
>
> **A5:** Thank you for this important suggestion. To make this part more complete, we will add more structured-sparsity results in the appendix. Specifically, we will include a 1.3B pre-training comparison (trained with 50B tokens) between block top-K and standard top-K at 50% sparsity,
>
> |Type|Sparsity|ARC-Easy|ARC-Challenge|HellaSwag|LAMBABA|PIQA|Winogrande|Avg.|
> |---|---:|---:|---:|---:|---:|---:|---:|---:|
> |Block TopK (size=32)|50%|45.5|26.2|44.5|47.6|66.9|52.1|47.13|
> |TopK|50%|47.0|25.5|45.1|47.9|67.2|52.0|47.45|
>
> |Type|Sparsity|C4 PPL|
> |---|---:|---:|
> |Block TopK (size=32) |50%|12.03|
> |TopK|50%|11.79|
>
> as well as training-free block top-K results on Qwen2.5-7B and Qwen2.5-32B under different block sizes and sparsity ratios.
>
> Qwen2.5-7B:
> |Type|Sparsity|ARC-C|HellaSwag|PIQA|Winogrande|MMLU|GSM-8K|MBPP| CommonsenseQA |Avg.|
> |---|---:|---:|---:|---:|---:|---:|---:|---:|---:|---:|
> |Dense|0%|51.6|79.0|79.8|73.1|71.9|82.8|64.2|85.4|73.47|
> |TopK|20%|51.6|78.8|79.8|72.9|71.5|83.4|64.4|85.0|73.43|
> |Block TopK (size=32)|20%|51.1|78.6|79.5|73.6|71.4|82.3|63.8|84.8|73.14|
> |Block TopK (size=128)|20%|51.2|78.7|79.5|72.9|71.3|82.9|63.6|84.8|73.11|
> |TopK|30%|51.5|78.4|79.3|71.1|71.1|80.9|62.6|84.0|72.36|
> |Block TopK (size=32)|30%|51.0|77.8|78.6|72.3|70.6|79.5|63.2|84.0|72.125|
> |Block TopK (size=128)|30%|51.0|78.0|78.7|72.7|70.5|81.3|62.8|84.1|72.39|
> |TopK|40%|52.0|77.4|79.0|71.2|69.7|78.8|61.6|83.3|71.63|
> |Block TopK (size=32)|40%|50.7|76.2|78.6|70.8|68.8|77.5|59.8|81.7|70.51|
> |Block TopK (size=128)|40%|50.4|76.6|78.8|71.0|69.4|76.1|58.6|81.4|70.29|
> |TopK|50%|50.3|74.9|78.5|71.0|67.7|71.6|53.8|80.3|68.51|
> |Block TopK (size=32)|50%|48.5|73.1|76.9|67.2|65.1|68.5|52.2|77.8|66.16|
> |Block TopK (size=128)|50%|48.3|73.8|78.0|67.8|65.9|68.5|53.2|80.0|66.94|
> |TopK|60%|49.2|69.0|77.0|65.9|61.5|54.2|43.2|72.6|61.58|
> |Block TopK (size=32)|60%|40.0|60.6|71.9|61.8|51.7|30.6|24.2|60.4|50.15|
> |Block TopK (size=128)|60%|44.2|65.8|74.2|61.5|57.8|45.9|33.6|66.3|56.16|
>
> Qwen2.5-32B:
> |Type|Sparsity|ARC-C|HellaSwag|PIQA|Winogrande|MMLU|GSM-8K|MBPP| CommonsenseQA |Avg.|
> |---|---:|---:|---:|---:|---:|---:|---:|---:|---:|---:|
> |Dense|0%|55.7|84.1|82.3|75.3|80.8|89.8|73.4|88.4|78.73|
> |TopK|20%|56.5|84.1|82.4|76.5|80.8|89.4|73.2|88.6|78.94|
> |Block TopK (size=32)|20%|57.0|84.0|82.1|74.2|80.7|89.8|72.8|87.9|78.56|
> |Block TopK (size=128)|20%|56.0|84.1|82.1|76.1|80.6|89.0|73.6|88.5|78.75|
> |TopK|30%|56.0|83.8|82.8|76.6|80.5|90.1|74.0|87.9|78.96|
> |Block TopK (size=32)|30%|56.6|83.6|82.0|75.5|80.4|90.4|74.6|87.6|78.84|
> |Block TopK (size=128)|30%|55.7|83.5|82.2|76.4|80.3|88.8|71.6|87.6|78.26|
> |TopK|40%|55.8|83.2|82.3|75.1|79.7|88.9|71.8|87.8|78.08|
> |Block TopK (size=32)|40%|55.1|82.5|81.2|75.6|79.4|89.5|70.4|86.7|77.55|
> |Block TopK (size=128)|40%|55.5|82.9|82.3|74.7|79.3|89.7|70.6|87.0|77.75|
> |TopK|50%|55.3|81.6|81.1|75.3|78.3|87.0|67.2|86.2|76.50|
> |Block TopK (size=32)|50%|54.0|80.9|81.6|73.3|77.2|86.1|68.0|83.6|75.59|
> |Block TopK (size=128)|50%|53.0|81.2|81.6|75.5|77.5|86.1|69.4|84.4|76.09|
> |TopK|60%|53.9|78.1|79.8|73.4|74.1|80.7|61.8|81.5|72.91|
> |Block TopK (size=32)|60%|50.3|73.3|76.8|70.8|68.5|71.1|47.4|75.6|66.73|
> |Block TopK (size=128)|60%|52.7|76.5|79.0|73.5|71.9|77.9|61.2|78.5|71.40|
>
>
> These results show that structured sparsification can achieve performance comparable to unstructured sparsification when training from scratch. However, under training-free sparsification, it exhibits more noticeable degradation, especially at smaller block sizes. We hope these additions will provide a broader structured-sparsity view and encourage further exploration in this direction.
>
> **Q6:** Tighten a few overstatements and wording issues.
>
> **A6:** Thank you for pointing this out. We agree that several statements can be made more precise. We have already revised some ambiguous wording based on Reviewer YPYh’s suggestions (see Reviewer YPYh, A3 and A4), and we will further improve the clarity, grammar, and presentation in the next revision. If there are additional specific examples, we would be happy to address them as well.
>
> [1] Liu, James, et al. "Training-free activation sparsity in large language models." ICLR 2025.

---

> ### Author Response · Authors · 2026-04-07
>
> Dear Reviewer 3Yir,
>
> I hope you are doing well.
>
> I am writing to follow up on the discussion, as there is about one week remaining before the discussion period ends, and I have not yet received any further response. I wanted to kindly check whether you might have any additional questions, comments, or concerns that I could help address.
>
> I would be very happy to provide any further clarification if needed. Thank you again for your time and consideration.
>
> Best regards,
>
> Authors.

---

> > ### Comment · Reviewer_3Yir · 2026-04-07
> > **Thanks for the clarification and follow-up**
> >
> > Dear Authors,
> >
> > Thank you for your polite follow-up and for your thoughtful engagement throughout the discussion.
> >
> > Your responses have been helpful, and at this stage I do not have further questions. I appreciate the clarifications you have provided and will consider them in my final assessment.

---

> > > ### Author Response · Authors · 2026-04-07
> > >
> > > We sincerely thank you for the thoughtful and constructive comments. We are very pleased to know that our responses have satisfactorily addressed your concerns. We greatly appreciate the time and effort invested in evaluating our work. Your feedback has been highly valuable and has helped improve the quality of our paper.

---

### Review · Reviewer_YPYh · 2026-03-19

**Summary Of Contributions:**

The authors observe that current approaches to sparsify LLMs are limited to only a small fraction of the MLPs used throughout the networks. They investigate the effect of training and postprocessing LLMs with fully sparse activations and do studies on the scaling behavior of their approach.

**Audience:**

Yes

**Audience Explanation:**

While the proposed approach is very nieche I would assume that researchers working on sparsifying neural networks would be interested in this kind of study.

**Claims And Evidence:**

Yes

**Claims Explanation:**

**Strengths**
- Easy to follow explanation of differences to previous work in 2.1
- Claims are followed by evidence in a structured way
- Every design decision is convincingly ablated
- With their approach, the authors ablate a design choice that has remained fixed in numerous previous works.

**Weaknesses**
- Limitations of non-structured sparsity and the single batch setting could be highlighted more
- I would have found an inference speed vs. performance plot for different sparsity ratios helpful. It is hard to match Table 1 and the other results
- What exactly is the purpose of the coloring in Tables 3,4,5...?

**Remarks**
- A block-sparsity structured version of the algorithm is introduced toward the end of the work, not in the results but in the discussion section. The approach appears to yield very similar results. Since structured sparsity has considerably fewer limitations, why did the authors decide to make this result a minor side note in their work?

**Requested Changes:**

Comments:
- Please clarify: In the introduction, "with dense activations, larger models remain more robust to aggressive activation sparsification." More robust than what? Small networks?
- Clarify in 4.2: "and a 700M model with 60% sparsity ratio achieves an improvement of 0.43% average accuracy" over what?
- Limitations of non-structured sparsity and the single batch setting should be highlighted more
- Since the structured sparsity approach performs similarly to the unstructured one. Why not make the structured sparsity approach the default algorithm? This needs to be discussed in the paper.

---

> ### Author Response · Authors · 2026-03-23
> **Part1**
>
> We thank the reviewer for the careful reading of our paper and for the constructive feedback.
>
> **Q1:** Limitations of non-structured sparsity and the single batch setting could be highlighted more. Why not make the structured sparsity approach the default algorithm?
>
> **A1:** Thank you for this important suggestion. **Our primary goal in this paper is to study the scaling behavior of fully sparse activations under the unstructured sparsity setting, since unstructured activation sparsification is the dominant setup in prior work on activation sparsity in LLMs [1,2,3].** Accordingly, we chose it as the main experimental protocol so that our results can be interpreted in direct continuity with the existing literature.
>
> We agree that structured sparsity is an important practical direction and deserves clearer discussion. Our intention was not to suggest that structured sparsity is less relevant, but rather to keep the main paper focused on the central scientific question: how model quality changes with model scale, token budget, and sparsity ratio under the standard unstructured setting used in prior literature.
>
> To address the concern more directly, we will add additional structured-sparsity results in the appendix. Specifically, beyond the existing 300M/700M-scale comparisons in the paper, we will include:
> * Pre-training comparison between block top-K (structured) and top-K (unstructured) sparsity at the 1.3B scale with 50% sparsity. Both models are trained with 50B tokens.
>
> |Type|Sparsity|ARC-Easy|ARC-Challenge|HellaSwag|LAMBABA|PIQA|Winogrande|Avg.|
> |---|---:|---:|---:|---:|---:|---:|---:|---:|
> |Block TopK (size=32)|50%|45.5|26.2|44.5|47.6|66.9|52.1|47.13|
> |TopK|50%|47.0|25.5|45.1|47.9|67.2|52.0|47.45|
>
> |Type|Sparsity|C4 PPL|
> |---|---:|---:|
> |Block TopK (size=32) |50%|12.03|
> |TopK|50%|11.79|
> * Training-free activation sparsification results on Qwen2.5-7B/32B under different block sizes (32/128) and sparsity ratios (0/20\%/30\%/40\%/50\%/60\%).
>
> Qwen2.5-7B:
> |Type|Sparsity|ARC-C|HellaSwag|PIQA|Winogrande|MMLU|GSM-8K|MBPP|CommonsenseQA|Avg.|
> |---|---:|---:|---:|---:|---:|---:|---:|---:|---:|---:|
> |Dense|0%|51.6|79.0|79.8|73.1|71.9|82.8|64.2|85.4|73.47|
> |TopK|20%|51.6|78.8|79.8|72.9|71.5|83.4|64.4|85.0|73.43|
> |Block TopK (size=32)|20%|51.1|78.6|79.5|73.6|71.4|82.3|63.8|84.8|73.14|
> |Block TopK (size=128)|20%|51.2|78.7|79.5|72.9|71.3|82.9|63.6|84.8|73.11|
> |TopK|30%|51.5|78.4|79.3|71.1|71.1|80.9|62.6|84.0|72.36|
> |Block TopK (size=32)|30%|51.0|77.8|78.6|72.3|70.6|79.5|63.2|84.0|72.13|
> |Block TopK (size=128)|30%|51.0|78.0|78.7|72.7|70.5|81.3|62.8|84.1|72.39|
> |TopK|40%|52.0|77.4|79.0|71.2|69.7|78.8|61.6|83.3|71.63|
> |Block TopK (size=32)|40%|50.7|76.2|78.6|70.8|68.8|77.5|59.8|81.7|70.51|
> |Block TopK (size=128)|40%|50.4|76.6|78.8|71.0|69.4|76.1|58.6|81.4|70.29|
> |TopK|50%|50.3|74.9|78.5|71.0|67.7|71.6|53.8|80.3|68.51|
> |Block TopK (size=32)|50%|48.5|73.1|76.9|67.2|65.1|68.5|52.2|77.8|66.16|
> |Block TopK (size=128)|50%|48.3|73.8|78.0|67.8|65.9|68.5|53.2|80.0|66.94|
> |TopK|60%|49.2|69.0|77.0|65.9|61.5|54.2|43.2|72.6|61.58|
> |Block TopK (size=32)|60%|40.0|60.6|71.9|61.8|51.7|30.6|24.2|60.4|50.15|
> |Block TopK (size=128)|60%|44.2|65.8|74.2|61.5|57.8|45.9|33.6|66.3|56.16|
>
> Qwen2.5-32B:
> |Type|Sparsity|ARC-C|HellaSwag|PIQA|Winogrande|MMLU|GSM-8K|MBPP|CommonsenseQA|Avg.|
> |---|---:|---:|---:|---:|---:|---:|---:|---:|---:|---:|
> |Dense|0%|55.7|84.1|82.3|75.3|80.8|89.8|73.4|88.4|78.73|
> |TopK|20%|56.5|84.1|82.4|76.5|80.8|89.4|73.2|88.6|78.94|
> |Block TopK (size=32)|20%|57.0|84.0|82.1|74.2|80.7|89.8|72.8|87.9|78.56|
> |Block TopK (size=128)|20%|56.0|84.1|82.1|76.1|80.6|89.0|73.6|88.5|78.75|
> |TopK|30%|56.0|83.8|82.8|76.6|80.5|90.1|74.0|87.9|78.96|
> |Block TopK (size=32)|30%|56.6|83.6|82.0|75.5|80.4|90.4|74.6|87.6|78.84|
> |Block TopK (size=128)|30%|55.7|83.5|82.2|76.4|80.3|88.8|71.6|87.6|78.26|
> |TopK|40%|55.8|83.2|82.3|75.1|79.7|88.9|71.8|87.8|78.08|
> |Block TopK (size=32)|40%|55.1|82.5|81.2|75.6|79.4|89.5|70.4|86.7|77.55|
> |Block TopK (size=128)|40%|55.5|82.9|82.3|74.7|79.3|89.7|70.6|87.0|77.75|
> |TopK|50%|55.3|81.6|81.1|75.3|78.3|87.0|67.2|86.2|76.50|
> |Block TopK (size=32)|50%|54.0|80.9|81.6|73.3|77.2|86.1|68.0|83.6|75.59|
> |Block TopK (size=128)|50%|53.0|81.2|81.6|75.5|77.5|86.1|69.4|84.4|76.09|
> |TopK|60%|53.9|78.1|79.8|73.4|74.1|80.7|61.8|81.5|72.91|
> |Block TopK (size=32)|60%|50.3|73.3|76.8|70.8|68.5|71.1|47.4|75.6|66.73|
> |Block TopK (size=128)|60%|52.7|76.5|79.0|73.5|71.9|77.9|61.2|78.5|71.40|
>
> These results show that structured sparsification can achieve performance comparable to unstructured sparsification when training from scratch. However, under training-free sparsification, it exhibits more noticeable degradation, especially at smaller block sizes. At the same time, we would still prefer to keep unstructured sparsity as the default setting in the main paper, because it is the most standard setting in prior activation-sparsity studies and therefore the most appropriate one for a systematic scaling-behavior study.

---

> ### Author Response · Authors · 2026-03-23
> **Part2**
>
> **Q2:** What exactly is the purpose of the coloring in Tables 3,4,5...?
>
> **A2:** Thank you for pointing this out. The coloring in Tables 3/4/5 is only for readability: it helps readers distinguish different sparsity levels (from darker to lighter shades) and more easily track trends across settings.
>
> **Q3:** Please clarify: In the introduction, "with dense activations, larger models remain more robust to aggressive activation sparsification." More robust than what? Small networks?
>
> **A3:** We agree this sentence is ambiguous. The intended meaning is that larger models are more robust than smaller models to aggressive activation sparsification. We will revise the introduction accordingly. This is consistent with our results showing that, at a fixed sparsity ratio, the gap to the dense baseline shrinks with model scale.
>
> **Q4:** Clarify in 4.2: "and a 700M model with 60% sparsity ratio achieves an improvement of 0.43% average accuracy" over what?
>
> **A4:** Thank you for catching this. The intended baseline is a dense 300M model: the sentence means that a 700M model with 60% sparsity achieves 0.43% higher average accuracy than a dense 300M model. We will rewrite this sentence to make the comparison explicit.
>
> [1] Liu, Zichang, et al. "Deja vu: Contextual sparsity for efficient llms at inference time." ICML 2023.
>
> [2] Liu, James, et al. "Training-free activation sparsity in large language models." ICLR 2025.
>
> [3] Mirzadeh, Seyed Iman, et al. "ReLU Strikes Back: Exploiting Activation Sparsity in Large Language Models." ICLR 2024.

---

> > ### Comment · Reviewer_YPYh · 2026-03-23
> > **Thanks for the response!**
> >
> > Dear authors,
> >
> > Q1: Thanks for the explanation. Given that most previous work focuses on unstructured sparsity, I agree that this is a suitable default setting for the paper story. The additional unstructured experiments provide further context for the interested reader.
> >
> > Q2: Maybe this could be highlighted with one sentence in the caption.
> >
> > Q3: Sounds good!
> >
> > Q4: Thanks!
> >
> > I will watch the discussion with the other reviewers. My main concerns have been addressed.

---

> > > ### Author Response · Authors · 2026-03-23
> > >
> > > We are glad that our response has addressed your concerns. We will incorporate the rebuttal content and the corresponding clarifications in the next revision.

---

### Decision · Action_Editor_ZsaC · 2026-04-22

**Recommendation:** Accept as is

**Audience:**

Yes

**Audience Explanation:**

LLMs (and other large foundation models) are widely used in ML and other scientific domains nowadays. Reducing model size and inference time is an ongoing and important challenge. This paper provides a solid academic work to help solve this challenge. I am certain the TMLR audience will find interest in such a work.

**Claims And Evidence:**

Yes

**Claims Explanation:**

The paper studies an important problem relevant to modern LLMs and other foundation models. Specifically, the authors aim to make the model more efficient by sparsifying the activations. The reviewers found the paper sound and  agreed that the paper is relevant to the TMLR audience. I agree that the paper's findings could be useful for many existing foundation models (beyond LLMs). During the review, the reviewers asked the authors to improve clarity regarding the efficiency claims and to strengthen the discussion of structured sparsity. I believe the authors did a good job and added new insights and results in the rebuttal.  All reviewers recommend acceptance of the paper in their final decision. Overall, this is a solid contribution and well-suited for TMLR. One clear strength of the work is the authors' systematic evaluation of the studied problem. I recommend acceptance of the paper.